# PTENβ is an alternatively translated isoform of PTEN that regulates rDNA transcription

Hui Liang[1,*], Xi Chen[1,*], Qi Yin[1], Danhui Ruan[1], Xuyang Zhao[1], Cong Zhang[1], Michael A. McNutt[1] & Yuxin Yin[1]

PTEN is a critical tumour suppressor that is frequently mutated in human cancer. We have previously identified a CUG initiated PTEN isoform designated PTENα, which functions in mitochondrial bioenergetics. Here we report the identification of another N-terminal extended PTEN isoform, designated PTENβ. PTENβ translation is initiated from an AUU codon upstream of and in-frame with the AUG initiation sequence for canonical PTEN. We show that the Kozak context and a downstream hairpin structure are critical for this alternative initiation. PTENβ localizes predominantly in the nucleolus, and physically associates with and dephosphorylates nucleolin, which is a multifunctional nucleolar phosphoprotein. Disruption of PTENβ alters rDNA transcription and promotes ribosomal biogenesis, and this effect can be reversed by re-introduction of PTENβ. Our data show that PTENβ regulates pre-rRNA synthesis and cellular proliferation. These results demonstrate the complexity of the PTEN protein family and the diversity of its functions.

[1] Institute of Systems Biomedicine, Department of Pathology, School of Basic Medicine, Beijing Key Laboratory of Tumor Systems Biology, Peking-Tsinghua Center of Life Sciences, Peking University Health Science Center, Beijing 100191, China. * These authors contributed equally to this work. Correspondence and requests for materials should be addressed to Y.Y. (email: yinyuxin@hsc.pku.edu.cn).

The PTEN gene was first identified and cloned in 1997, and PTEN protein has proved to be a powerful tumour suppressor[1,2]. It is one of the most frequently mutated or deleted genes in cancer, including cancer of the brain, breast, prostate, pancreas and ovary[3]. Germline mutations of the PTEN gene lead to inherited autosomal dominant hamartoma tumour syndromes, including Bannayan–Zonana syndrome[4] and Cowden disease[5]. Heterozygous Pten deletion leads to multiple tumours in mice, while the homozygote is embryonically lethal[6,7]. In the cytoplasm, PTEN antagonizes the PI3K-AKT-mTOR pathway by dephosphorylation of phosphoinositide-3,4,5-triphosphate (PIP3)[8]. Loss of PTEN function leads to uncontrolled activation of the PI3K/AKT pathway, which stimulates cell growth, cell proliferation and cell survival, and inhibits apoptosis. PTEN also accumulates in the cell nucleus through passive diffusion, Ran-MVP-mediated transport and mono-ubiquitination-regulated mechanisms[9]. In the nucleus, PTEN maintains genome integrity and regulates the process of DNA replication through regulation of RPA1, MCM2 and TOP2A (refs 10–13). In addition to its tumour suppressor function, PTEN is also involved in embryonic development, lipid metabolism[14,15], Alzheimer's disease[16] and antiviral innate immunity[17]. However, the current concepts of PTEN function do not fully explain the multifarious activity of this protein.

In eukaryotes, the translation initiation codon is generally recognized and selected by the 5′cap structure dependent ribosomal scanning mechanism, as well as by 5′cap structure independent internal ribosome entry sites. This scanning complex rigorously controls the fidelity of initiation through recognition of the correct AUG triplet in the optimum context GCC(A/G)CCAUGG, referred to as the Kozak sequence, which includes a purine at the $-3$ and a G at the $+4$ position (relative to the A of the AUG codon, which is designated $+1$) (refs 18,19).

Triplets that differ from AUG by only one nucleotide can also direct initiation of polypeptide chain synthesis in mammalian cells in an optimum context[20]. It is also reported that recognition of non-AUG codons is strongly stimulated by a downstream stem-loop (hairpin) structure that is separated from the preceding initiation codon by about 14 nucleotides[21]. Isoforms initiated from non-AUG codons are frequently endowed with functions additional to and differing from the canonical form of a given protein[22,23].

Previously, we and others reported that a CUG codon in the 5′ untranslated region (5′ UTR) of PTEN mRNA initiates an N-terminal extended isoform of PTEN now known as PTENα (refs 24,25). In the current study we describe another PTEN isoform designated PTENβ, which comprises an extended N-terminal extension of 146 amino acids (Homo sapiens) as compared with canonical PTEN protein. Our data show PTENβ localizes specifically in the nucleolus and negatively regulates rDNA transcription. Disruption of PTENβ leads to an abnormal increase in pre-rRNA synthesis and results in promotion of cellular proliferation. These findings characterize an N-terminal extended PTEN isoform with distinct localization and function, which results from alternative AUU translation initiation. Identification of PTENβ further reveals the diversity of PTEN protein family function, and the correspondence of protein function with isoform subtype.

## Results

### An unidentified protein is recognized by a PTENα antibody.
The protein PTENα functions in mitochondrial metabolism, and structurally distinct compounds such as acriflavin and aurin tricarboxylic acid (ATA) yield different results in regulation of PTENα synthesis initiated by a CUG start codon[25]. The CUG inhibitor acriflavin reduces PTENα expression in a dose-dependent manner without affecting canonical PTEN expression. We observed that a full length PTEN antibody reacted with an unidentified protein of molecular weight lower than PTENα (around 70 kDa), and expression of this protein showed dose dependent variation under acriflavin treatment parallel to that of PTENα (Fig. 1a). A panel of cancer cell lines was examined to determine whether differences in PTEN status affect this unidentified PTEN-like protein, and the protein was undetectable in PTEN-null PC3 cells (Fig. 1b). This protein was also recognized by a monoclonal antibody against the PTEN C-terminal domain, suggesting its C-terminal region is similar to or identical with canonical PTEN. In order to confirm this, we examined cancer cell lines with an anti-PTENα antibody raised in our laboratory by immunization with purified recombinant protein composed of the 173 N-terminal amino acids of PTENα (ref. 25). This unidentified protein was recognized by this anti-PTENα antibody (Fig. 1c), indicating that it was most likely another N-terminally extended PTEN variant generated by initiation at an in-frame initiation codon within the PTEN mRNA 5′ leader. We designated this PTEN-like protein PTENβ.

### Alternative initiation of PTENβ translation from AUU[594].
Evaluation of the 5′ UTR of human PTEN mRNA for alternative translation start sites in frame with the AUG[1032] start codon revealed a total of four non-AUG alternative initiation codons in favourable Kozak contexts (Fig. 2a). Translation from AUU[594] or UUG[621] is expected to encode larger forms of PTEN of 549 or 540 amino acids respectively, with predicted molecular weights of 61.6–62.9 kDa. A protein of this size should migrate under gel electrophoresis at around 70 kDa, matching the observed PTENβ band.

To determine whether one of these non-AUG codons initiates translation of this putative PTENβ protein with upstream extension of the open reading frame (ORF), the coding sequence of PTENα was cloned into a plasmid with a C-terminal GFP tag (Fig. 2b, upper panel). As expected, the CUG[513] initiated PTENα ORF was translated into three distinct proteins with masses of $120^+$ kDa (PTENα-GFP), 120 kDa (PTENβ-GFP) and 90 kDa (PTEN-GFP) (Fig. 2b, lower panel). This suggested that PTENβ is translated from a new ORF in the 5′ UTR of PTEN mRNA upstream of the canonical AUG start codon.

To identify PTENβ translation initiation sites, a plasmid was constructed to express CUG[513]-initiated PTENα with a C-terminal GFP tag, and mutations were created in this plasmid to determine which non-AUG codon is critical for PTENβ expression (Fig. 2c). Mutation of CUG[513] into a non-initiating CUC triplet led only to disappearance of the most slowly migrating proteoform PTENα, thus excluding the possibility that PTENβ is a cleavage product (Fig. 2d, lane 4). On the basis of CUG[513] mutation, potential non-AUG codons (AUU[594], UUG[621]) were mutated separately to a non-initiating CUC triplet. Anti-GFP immunoblots from these transfected lysates revealed that mutation of AUU[594] abolishes PTENβ, whereas mutation of UUG[621] has no such effect (Fig. 2d, lane 6 versus lane 5). These results indicate that AUU[594] but not UUG[621] is essential for PTENβ expression, and that AUU[594] is likely the translation initiation site for PTENβ.

AUU[594] resides in a favourable Kozak context as noted (GUCACCAUU[594]) (Fig. 2a). In order to determine whether this context influences translation initiation at AUU[594], we constructed mutants to disrupt it and examined PTENβ expression (Fig. 2e). Disruption of this Kozak context markedly

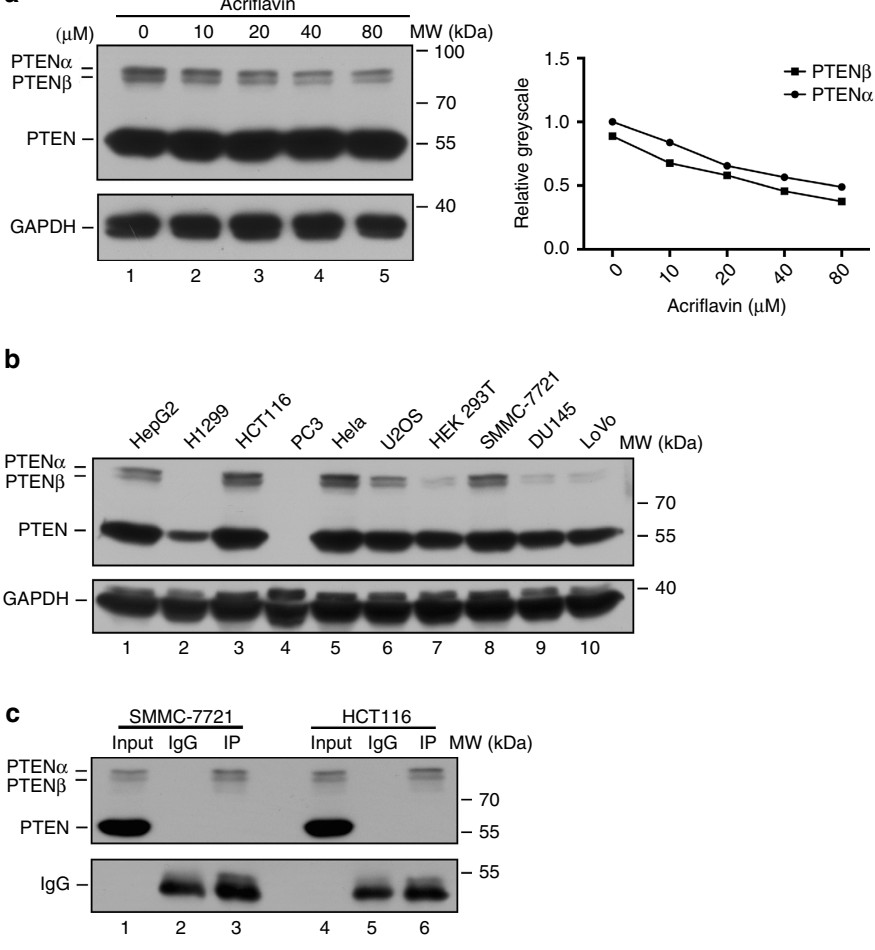

**Figure 1 | Discovery of an unknown protein that is recognized by PTENα antibody. (a)** Western blot of PTENα and PTEN in Hela cells after acriflavin treatment at different doses with a polyclonal antibody against full-length PTEN. PTEN immunoblotting revealed an unknown band with a molecular weight lower than PTENα. Quantification of indicated protein levels relative to GAPDH is shown (right panel). **(b)** Western blot of PTENα and PTEN in a panel of human cancer cell lines using a rabbit monoclonal antibody against the C-terminal region of PTEN (Cell Signaling, 138G6). The PTEN null prostate cancer cell line PC3 was used as a negative control. **(c)** Immunoprecipitation of PTENα with a homemade PTENα antibody before PTEN immunoblotting of SMMC-7721 and HCT116 cells of known PTEN status.

reduced PTENβ expression (Fig. 2f), strongly suggesting that translation initiation of PTENβ is reliant on the Kozak context.

Detailed evaluation of the 5′ UTR of human PTEN mRNA revealed a 12 bp perfect palindromic sequence starting 18 bp downstream of $AUU^{594}$, and phylogenetic analysis suggests that this secondary structure is evolutionarily conserved (Fig. 2g). As shown in Fig. 2h, disruption of this palindromic motif by mutation led to a marked reduction of PTENβ without affecting canonical PTEN expression (lower panel, lane 4 versus lane 3), indicating this sequence is critical for translation initiation of PTENβ.

These data demonstrate that PTENβ synthesis relies on an alternative translation initiation at the $AUU^{594}$ codon, and expression of PTENβ is influenced both by the Kozak context and by the secondary structure downstream of $AUU^{594}$.

**Mass spectrometry analysis of the PTENβ sequence.** Mass spectrometry was used for peptide sequencing to validate the PTENβ translation initiation site. Purified human PTENβ with a C-terminal His tag (Fig. 3a) was expressed in Sf9 insect cells for mass spectrometry (Fig. 3b). Mascot reports revealed three peptide fragments covering 39.04% of the N-terminal region of

PTENβ (from $AUU^{594}$ to $AUG^{1032}$) (Fig. 3b). LC-MS/MS captured MSRAGNAGE, which is the most proximal N-terminal peptide of PTENβ (9 aa, MS/MS spectrum shown in Fig. 3c). Additional analysis showed that the first N-terminal amino acid of PTENβ is acetylated. These mass spectrum data therefore verified $AUU^{594}$ is the PTENβ translation initiation site.

The 5′ UTR of *Homo sapiens PTEN* and *Mus musculus Pten* are highly homologous as shown in our recent study[25], suggesting that PTENβ expression in mice is conserved and has a similar alternative translation initiation mechanism. We previously generated a FLAG-knockin mouse model ($Pten^{FLAG}$), in which the C-terminus of the *Pten* gene was targeted for insertion of a FLAG-coding sequence[25]. To verify PTENβ resides at the *Pten* gene locus, tissue samples were extracted from heterozygous $Pten^{FLAG}$ mice and wild-type control mice for FLAG pull-down followed by immunoblotting with an anti-PTENα antibody raised in our laboratory[25]. A protein band with a molecular weight lower than PTENα which was similar to the PTENβ band identified in cancer cell lines was detected in the FLAG elute from $Pten^{FLAG}$ tissues, while no band was detected in wild-type tissues (Fig. 3d). The existence of *in vivo* PTENβ was further confirmed by mass spectrometric analysis of a FLAG-purified 70$^{+}$ kDa band (molecular weight

lower than PTENα) from *Pten^FLAG* tissues. Mascot reports revealed four peptide fragments that cover 68.5% of the N-terminal region of PTENβ (Fig. 3e). These results verified the existence of PTENβ *in vivo*, and confirmed that PTENβ and

PTENα are translated *in vivo* from identical mRNA at the same gene locus. This *Pten^FLAG* knockin animal model demonstrates the natural occurrence of alternative initiation that results in translation of PTENβ.

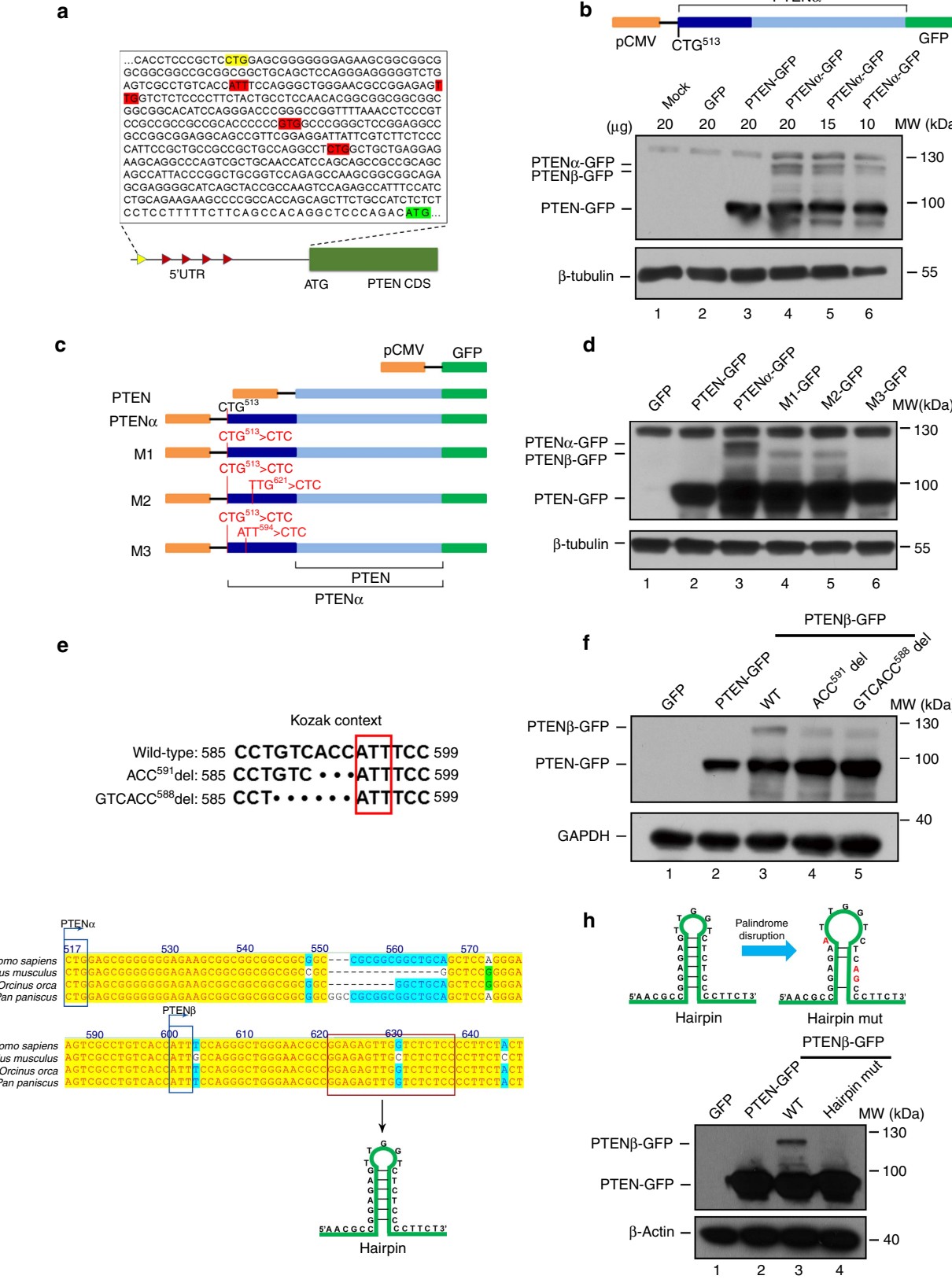

**PTENβ is localized predominantly in the nucleolus.** Longer protein isoforms resulting from alternative translation initiation frequently contain a signal for subcellular localization that is absent in shorter forms[26,27]. The length of the PTENβ protein differs from PTENα and PTEN due to its N-terminal extension, which raised the possibility that PTENβ has unique subcellular localization and special function. Immunofluorescence examination of the subcellular localization of PTENβ was compared with PTENα and PTEN. Separate plasmids expressing C-terminal GFP-tagged PTEN, PTENα or PTENβ were constructed, and to ensure expression of only a single isoform in each instance, we created an $ATG^{1032} > ATA$ mutation in GFP-tagged PTENα and PTENβ (Fig. 4a). The subcellular distribution of PTENβ was completely different from that of PTEN and PTENα as shown in Fig. 4b. PTEN was distributed ubiquitously in both the nucleus and cytoplasm, while PTENα predominately showed cytoplasmic localization. In contrast, strong PTENβ fluorescence signals were distributed mainly in the relatively shallow DAPI staining areas of the nucleus which have been suggested to be nucleolar regions[28], and little cytoplasmic PTENβ localization was detected. In order to exclude the possibility protein localization was influenced by the GFP tag, PTEN, PTENα and PTENβ expression plasmids were constructed without a tag (Supplementary Fig. 1a), and immunofluorescence with PTEN monoclonal antibody in PTEN null cells revealed that these untagged isoforms have subcellular distribution patterns identical to the GFP-tagged molecules (Supplementary Fig. 1b,c). UBF that is a well-known nucleolar marker was used to evaluate for co-localization of C-terminal GFP tagged PTENα, PTENβ or PTEN within the nucleolus. There was prominent co-localization of PTENβ with this marker in the nucleolus, but considerably less nucleolar co-localization was detected with PTEN or PTENα (Fig. 4c). This extensive distribution of PTENβ in the nucleolus was confirmed through cell fractionation (Fig. 4d). Moreover, we demonstrated that the N-terminal extended sequence was solely responsible for PTENβ accumulation in the nucleolus, and this was also true for PTENα localization on mitochondria as we reported previously (Supplementary Fig. 2).

The nucleolar-localization signals (NoLSs) that govern nucleolar localization and retention are usually rich in arginine and lysine residues[29,30]. We noted a series of poly-arginine residues in the N-terminal extended domain of PTENβ (aa 20–25), which is evolutionarily conserved (Supplementary Fig. 3). To determine whether this sequence acts as an NoLS and directs PTENβ nucleolar localization, we established a PTENβΔR$^6$ construct in which these six arginines were deleted. This ΔR$^6$ mutation was remarkable in that it completely abolished PTENβ nucleolar accumulation (Fig. 4e), indicating that this poly-arginine sequence is critical for PTENβ nucleolar localization, and

showing that it functions as an NoLS. These results indicated that the distinctive N-terminal extension sequence of PTENβ confers unique cellular localization, and raised the possibility that PTENβ is involved in nucleolar function.

**Nucleolin is a novel downstream target of PTENβ.** In order to identify potential PTENβ targets and explore PTENβ function in the nucleolus, mass spectrometry was used to analyse the results of S-tag-PTENβ pull down. Results with S-tag-PTEN and S-tag-PTENα were analysed separately as controls. Mass spectrometry revealed that PTENβ specifically interacts with a large number of nucleolar proteins (Supplementary Fig. 4a), including nucleolin that is the most abundant multifunctional phosphoprotein of the nucleolus. Nucleolin is comprehensively involved in ribosome biogenesis[31,32], and was thus considered to be a potential PTENβ target (Fig. 5a). Interaction of PTENβ and nucleolin was demonstrated by reciprocal immunoprecipitation (Fig. 5b,c). In addition, immunofluorescence showed extensive co-localization of PTENβ with nucleolin in the nucleolus, whereas PTEN or PTENα did not (Fig. 5d), further confirming the specificity of the interaction of PTENβ and nucleolin. To identify the functional domains by which PTENβ binds to nucleolin, two truncated forms of PTENβ were constructed with C-terminal S-tags (Supplementary Fig. 5a). Immunoprecipitation demonstrated that the N-terminal extension domain of PTENβ is essential for interaction with nucleolin (Supplementary Fig. 5b). These data show PTENβ and nucleolin co-localize and interact each other.

Like canonical PTEN, PTENβ contains an intact phosphatase domain, and we found that exogenous expression of PTENβ, PTENα or PTEN in PTEN null cells all efficiently reduce p-AKT to a similar level (Supplementary Fig. 6a). Moreover, missense mutations in the phosphatase domain of PTENβ (G275E, analogous to PTEN (G129E)) and (C270S, analogous to PTEN (C124S)) both eliminated the lipid phosphatase activity of the protein (Supplementary Fig. 6b). PTENβ thus has phosphatase activity that is presumably similar to PTENα and canonical PTEN. In addition, we found that affinity for nucleolin binding was largely diminished in protein phosphatase activity deficient PTENβ mutants (Y284L and C270S) (Fig. 5e, lane 3 and lane 5 versus lane 2 and lane 4), suggesting that PTENβ protein phosphatase activity is necessary for interaction with nucleolin. This further indicated PTENβ acts as a nucleolin phosphatase.

In order to clarify the relationship of PTENβ with nucleolin and evaluate PTENβ function in the nucleolus, somatic PTENβ knockout was established with CRISPR-Cas9. CRISPR-Cas9-mediated gene targeting eliminated PTENα and PTENβ without affecting canonical PTEN expression, and we also designed a specific target sequence for sole knockout of PTENα as a control

**Figure 2 | Identification and validation of PTENβ.** (**a**) The potential non-AUG initiation codons in favourable Kozak contexts in the 5′ UTR region of *Homo sapiens* PTEN mRNA are highlighted in red. The initiation codons of PTENα and PTEN are separately highlighted in yellow or green. (**b**) A pEGFP-N1 plasmid containing PTENα with a C-terminal GFP tag was used for detection of PTENα, PTEN and unknown PTEN isoforms (upper panel); the indicated plasmids were introduced in HEK 293T cells followed by western blotting analysis using GFP antibody. β-tubulin was used as a control (lower panel). (**c**) pEGFP-N1 plasmids containing PTEN or PTENα with a C-terminal GFP tag, in which one of the two potential initiation codons (AUU$^{594}$ or UUG$^{621}$) was mutated to CUC combined with mutation of CUG$^{513}$. (**d**) Mutation of AUU$^{594}$ but not UUG$^{621}$ eliminates PTENβ expression. Plasmids as indicated in **c** were introduced into HEK 293T cells separately, followed by immunoblotting with GFP antibody. (**e**) The Kozak context of AUU$^{594}$ and disruption of the Kozak context by site directed mutagenesis. (**f**) The Kozak context disruption abolishes PTENβ expression. HEK 293T cells were transfected with indicated constructs in **e**, followed by immunoblotting with GFP antibody. (**g**) A 12 bp AUU$^{594}$ downstream palindromic motif in the 5′ UTR of PTEN is evolutionarily conserved. Phylogenetic analysis of the 5′ UTR of PTEN mRNA in bonobo (*Pan paniscus*), killer whale (*Orcinus orca*) and mouse (*Mus musculus*). The alternative CUG$^{513}$ and AUU$^{594}$ codons are highlighted in blue boxes, and the 12 bp palindromic sequence is highlighted in a red box. (**h**) Disruption of the palindromic motif abolishes PTENβ expression. Upper panel, site directed mutagnesis was used to disrupt the AUU$^{594}$ downstream palindromic motif; lower panel, C-terminal GFP-tagged PTENβ expression plasmids with or without disruption of the palindromic motif were introduced into HEK 293T cells, followed by immunoblotting with GFP antibody.

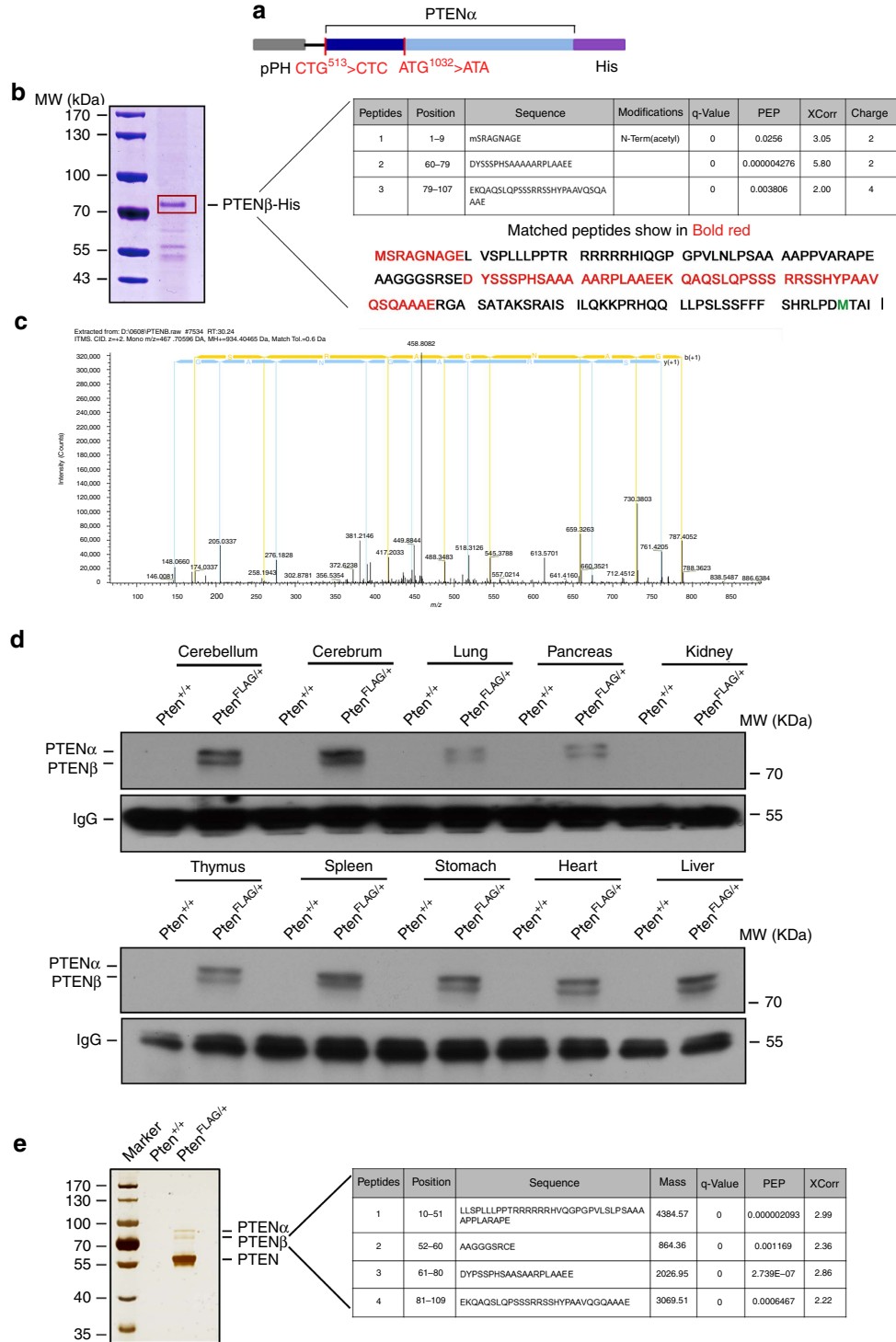

**Figure 3 | The PTENβ translation initiation codon was identified by LC-MS/MS.** (**a**) A pFastBac1 plasmid containing PTENα with a C-terminal His-tag was used for *in vitro* purification of PTENβ. The PTEN AUG start codon was mutated to AUA and the PTENα CUG start codon was mutated to CUC to avoid co-purification of PTEN and PTENα with PTENβ. (**b**) Mass spectrometry analysis of purified PTENβ. Sf9-expressed PTENβ with a C-terminal His-tag was purified using nickel affinity chromatography. The bound proteins were separated with SDS–PAGE, and gel slices at around 70 kDa (red box) were analysed by mass spectrometry. Three segments of peptide that match the 5′ UTR region of PTEN were identified, including the most proximal N-terminal peptide of PTENβ MSRAGNAGE. (**c**) The MS/MS spectrum of the most proximal N-terminal peptide of PTENβ MSRAGNAGE. (**d**) Verification of endogenous PTENβ expression in *Pten*-FLAG knockin mice. Various tissues from *Pten*-FLAG knockin mice were lysed for immunoprecipitation with FLAG antibody, followed by immunoblotting with PTENα antibody raised in our laboratory. Wild-type mice tissues were used as control. (**e**) Brain tissues from *Pten*-FLAG knockin mice or control wild-type mice were lysed for immunoprecipitation with anti-FLAG M2 agarose. The bound proteins were separated with SDS–PAGE. Mass spectrometry analysis of gel slices at around 70 kDa (molecular weight lower than PTENα) revealed four peptides in *Pten*-FLAG knockin tissues that matched the PTENβ N-terminal extended region.

(Fig. 5f,g). We found that PTENα and PTENβ double knockout led to elevation of nucleolin phosphorylation levels at Thr84, which is one of the XTPXKKXX motifs located in the nucleolin N-terminal region important for nucleolin function (Fig. 5h). Moreover, exogenous expression of wild-type PTENβ but not protein phosphatase activity abolished PTENβ efficiently

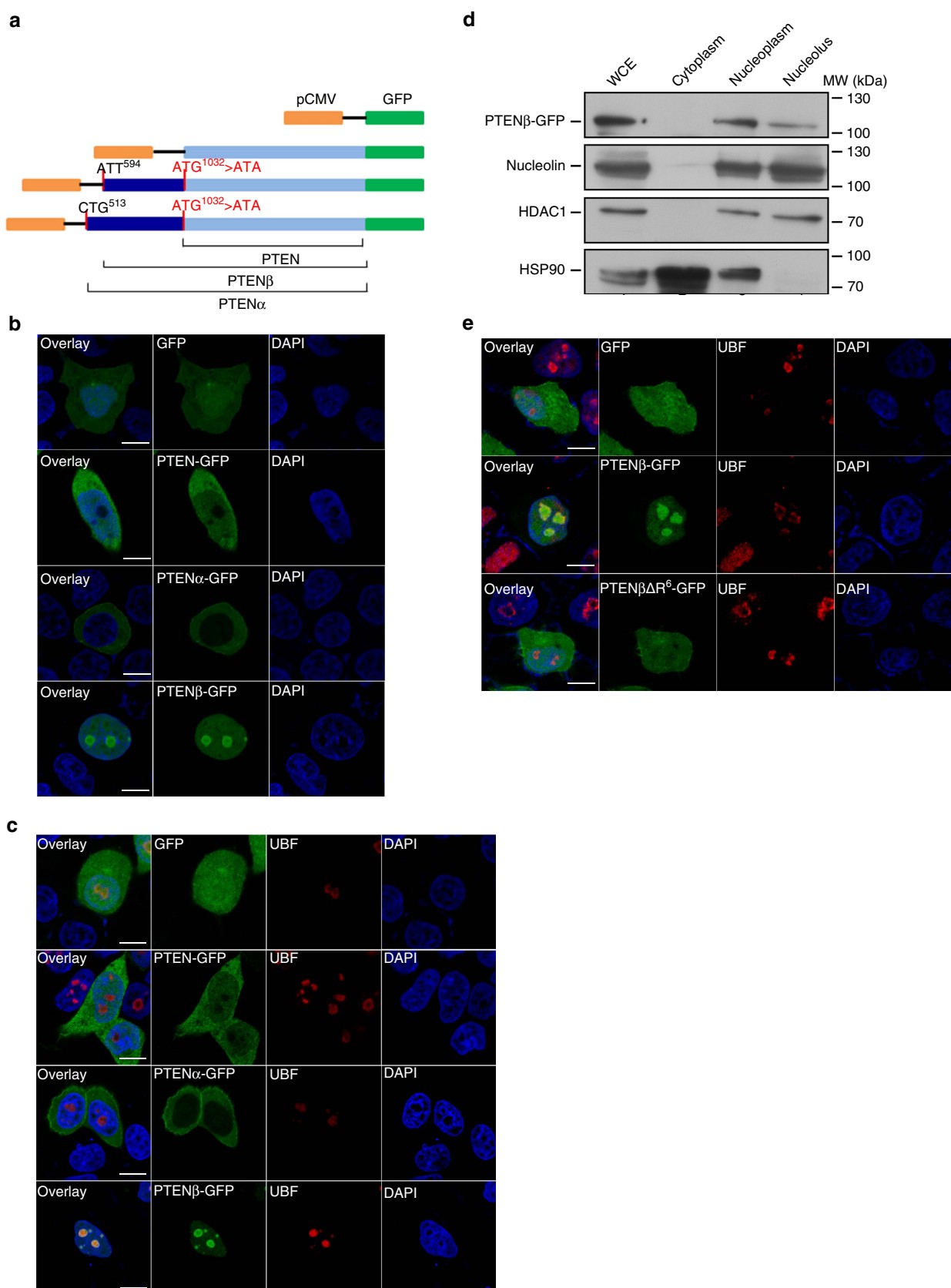

downregulated nucleolin phosphorylation levels at Thr84 (Fig. 5i). These results raised the possibility that PTENβ acts as a nucleolin phosphatase. *In vitro* dephosphorylation assays further verified Cdk1/CyclinB1-phosphorylated nucleolin Thr84 is a target of PTENβ for dephosphorylation (Fig. 5j, lane 5 versus lane 4). In addition, we found that the poly-arginine sequence deleted mutant that eliminated PTENβ nucleolar localization also effectively abolishes its phosphatase activity on nucleolin. This indicates that PTENβ nucleolin phosphatase activity is linked to its localization in the nucleolus (Supplementary Fig. 7). These results argue that PTENβ acts as a nucleolin phosphatase, and show that the PTENβ–nucleolin interaction results in dephosphorylation modification of nucleolin, which influences its function.

**PTENβ inhibits rDNA transcription and cell proliferation.** Nucleolin is involved in almost every step of ribosome biogenesis[33–35], and a recent study showed the most prominent effect of nucleolin is regulation of rDNA transcription by promotion of the rDNA transcription complex assembly, in which the phosphorylation status of nucleolin is rigorously regulated[36,37]. The fact that PTENβ interacts with nucleolin and dephosphorylates nucleolin at Thr84 raised a question as to whether nucleolin mediates PTENβ regulation of rDNA transcription, and thus represents a direct mechanism by which PTENβ controls ribosomal biogenesis.

To determine the influence of PTENβ on rRNA synthesis, total rRNA levels were evaluated by gel electrophoresis. Knockout of PTENα and PTENβ without affecting PTEN expression led to elevation of 28S and 18S total rRNA levels, whereas knockout of PTENα alone had no such effect (Fig. 6a, lane 3 versus lane 2 and lane 1). Moreover, exogenous expression of PTENβ but not PTENα in PTENα and PTENβ null cells efficiently downregulated 28S and 18S rRNA levels (Fig. 6b, lane 3 versus lane 2 and Supplementary Fig. 8a). These results argue that PTENβ plays a role in regulation of rRNA, and in particular is involved in regulation of pre-rRNA synthesis from the outset of ribosome biogenesis. Transcription levels of rRNA genes were therefore evaluated by analysing pre-rRNA with qPCR in PTENβ wild-type and knockout cells. As expected, 45S pre-rRNA was increased in PTENβ and PTENα double knockout cells as compared with PTENα only knockout or wild-type cells (Fig. 6c). To determine whether rDNA transcription is regulated by PTENβ, a pHrD-IRES-Luc or pGL3-basic control vector was transfected into PTENβ and PTENα double knockout cells, and the luciferase reporter activity of the rDNA promoter was measured. Introduction of PTENβ, but not PTENα, efficiently downregulated luciferase activity (Fig. 6d and Supplementary Fig. 8b). These results indicate that PTENβ negatively regulates rDNA transcription, which may act at least in part through PTENβ dephosphorylation of nucleolin.

Nucleolar morphology is closely related to ribosome biogenesis, and enlarged nucleoli are typically found in association with activation of ribosome biogenesis. To evaluate alterations in nucleolar morphology, silver staining was used for the identification of argyrophilic nucleolar organizer regions (AgNOR). PTENβ and PTENα double knockout resulted in greater average areas of AgNOR staining, while PTENα only knockout did not have this effect (Fig. 6e). This suggested that elimination of PTENβ leads to upregulation of ribosomal biogenesis. Stimulation of cell proliferation is a characteristic biologic feature of nucleolin, and is typically associated with elevated ribosomal biogenesis. We therefore investigated biologic changes resulting from regulation of ribosomal biogenesis by PTENβ. Growth curves showed significant increases in cell growth rate resulting from PTENβ knockout (Fig. 6f). This activation of cell proliferation resulting from PTENβ deprivation was further confirmed by S phase analysis (Fig. 6g). These results argue that PTENβ controls cellular proliferation and may act through a mechanism involving negative regulation of pre-rRNA synthesis.

## Discussion

In eukaryotic cells, the ribosome typically initiates protein synthesis at the AUG codon closest to the 5' end of the mRNA [38]. However, there is now evidence including observations from our previous study[25] that demonstrates initiation of protein translation can occur at most codons which differ from AUG by no more than a single nucleotide (non-AUG initiation)[20]. More than 50 instances of non-AUG-initiated N-terminal extensions have been predicted or verified experimentally in mammals[39,40]. In this study, we report identification of a translational variant of PTEN, designated PTENβ. This protein is synthesized from an alternative translation start site 438 bp upstream of the AUG initiation sequence, which adds 146 N-terminal amino acids to the canonical PTEN ORF. We found that PTENβ specifically localizes in the nucleolus and inhibits pre-rRNA synthesis, and our experimental evidence strongly suggests this inhibition acts through the mechanism of nucleolin dephosphorylation.

In previous studies both our group and Hopkins *et al.* independently identified a translational variant of PTEN with N-terminal extension, designated 'PTENα' in our study, and 'PTEN-long' by Hopkins *et al.*[24]. These have proven to be the same isoform that is translated from a CUG codon 519 bp upstream of the AUG initiation sequence of canonical PTEN (refs 24,25). In addition to the initiation sites for PTENα and PTENβ (CUG[513], AUU[594]), we found another two alternative non-AUG codons in favourable Kozak contexts, which may be sites for initiation of other as yet unidentified isoforms. As multiple additional forms of PTEN may exist, we consider it to be prudent to designate PTEN isoforms as a sequential series, using α, β, γ, δ as labels.

In eukaryotes, the efficiency of initiation depends on the Kozak context[41] which encompasses the AUG codon. Our data show that initiation of PTENβ relies on the Kozak context surrounding AUU[594], which suggests this context is also essential for efficient initiation of other non-AUG codons. We identified a 12 bp perfect palindromic sequence starting 18 bp downstream of AUU[594], and showed that disruption of this conserved

---

**Figure 4 | PTENβ is localized predominantly in the nucleolus.** (**a**) A set of different constructs of PTEN, PTENα and PTENβ with a C-terminal GFP tag. The AUG start codon of canonical PTEN was mutated to AUA in PTENα and PTENβ constructs in order to abolish expression of canonical PTEN. (**b**) Subcellular localization of C-terminal GFP-tagged PTENα, PTENβ and PTEN. The constructs indicated in **a** were introduced into Hela PTEN null cells. Twenty-four hours after transfection, cells were stained with DAPI, followed by imaging with confocal microscopy. The scale bars represent 5 μm. (**c**) Hela PTEN null cells were transfected as in **b**, then stained with an anti-UBF antibody and DAPI, and imaged with confocal microscopy. The scale bars represent 5 μm. (**d**) Hela cells were transfected with C-terminal GFP-tagged PTENβ, and subjected to cell fractionation, followed by western blotting with GFP, nucleolin, HDAC1 and HSP90 antibodies. WCE: whole cell lysate. (**e**) Hela PTEN null cells were transfected with C-terminal GFP-tagged PTENβ or C-terminal GFP-tagged PTENβΔR[6]. Twenty-four hours after transfection, cells were stained with anti-UBF antibody and DAPI, followed by imaging with confocal microscopy. The scale bars represent 5 μm.

palindrome sharply downregulates expression of PTENβ. The crucial role this downstream palindromic sequence plays in AUU-mediated initiation, together with discovery of the fact in our previous study that the CUG centred palindromic sequence may be a signature motif for alternative Leu-tRNA initiation[25], underscores the importance of secondary structure for non-AUG initiation. This is consistent with the concept that a strong secondary structure downstream of the non-AUG codon may significantly increase initiation efficiency as reported by Kozak[42].

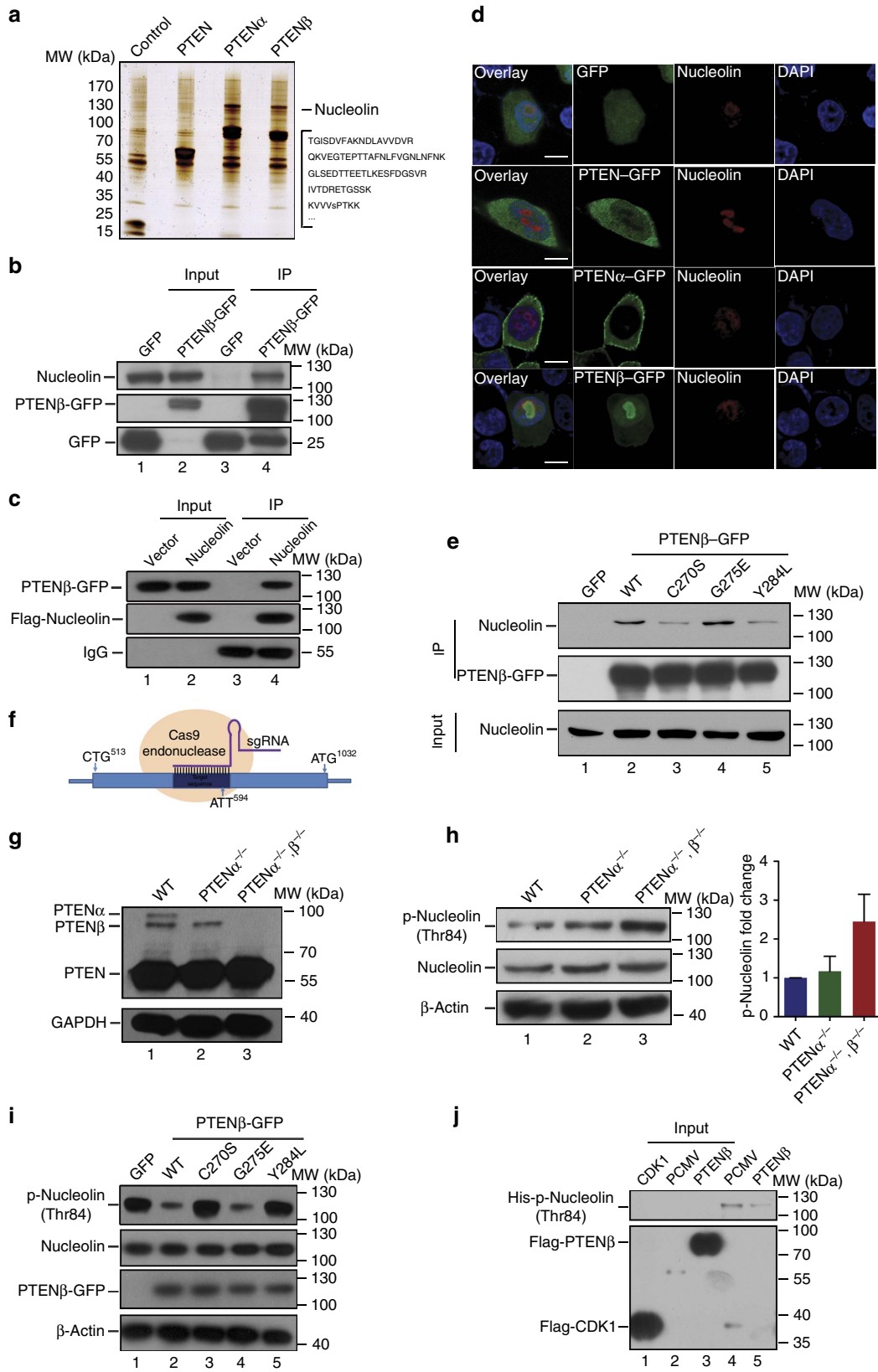

We previously showed that PTENα localizes in mitochondria and functions in mitochondrial metabolism[25]. Hopkins et al. reported that PTEN-long (designated PTEN-L) is secreted into adjacent cells, which relies on an N-terminal secretion sequence[24]. These findings indicate that the N-terminal extended PTENα isoform is endowed with capability for subcellular localization differing from canonical PTEN. Data in the current study show that there is partial co-localization of PTEN and PTENα in the cytoplasm, and this is consistent with findings in our previous study that PTEN interacts and co-localizes with PTENα in mitochondria[25]. However, PTEN isoforms localize for the most part in a mutually exclusive manner (Supplementary Fig. 9). PTENβ specifically localizes in the nucleolus, whereas PTEN and PTENα do not, and an evolutionarily conserved poly-arginine sequence has proven to be an NoLS of PTENβ. The PTENβ N-terminus is 27 amino acids shorter than PTENα, and PTENβ therefore does not contain the N-terminal secretion signal sequence, which may in part account for the differences in cellular distribution of PTENβ and PTENα. Moreover, protein conformation may be altered by differences in the length of the N-terminal domain, leading to differing affinities for interacting proteins or nucleotide sequences[27,43]. This may in some measure explain the diversity of PTEN isoform distribution.

Our data demonstrate that PTENβ is involved in regulation of pre-rRNA synthesis, and disruption of PTENβ results in elevated ribosomal biogenesis and cellular proliferation. Our findings suggest that this PTENβ regulatory mechanism may act through dephosphorylation of nucleolar protein nucleolin. It has been reported that PTEN acts on ribosomal biogenesis through the PI3K-mTOR pathway by downregulation of 45S pre-rRNA and that PTEN localizes to the nucleolus to regulate nucleolar homeostasis[44–46]. However, our results indicate there is extensive distribution of PTENβ in the nucleolus while little nucleolar PTEN can be detected. The PTENβ coding sequence overlaps the sequence of full-length canonical PTEN and PTENβ can be efficiently recognized by PTEN antibodies. It is therefore possible that the nucleolar immunofluorescence signals detected by the PTEN monoclonal antibody in this study are derived from recognition of this previously unidentified PTENβ isoform instead of from canonical PTEN. Furthermore, it must be noted that most of the observations from the studies cited above were made in PTEN knockout cells or with Pten deficient mice. As PTEN and its isoforms PTENα and PTENβ are coded by the same gene, current PTEN knockout mouse models are essentially equivalent to at least triple deletion of PTEN and its isoforms. The phenotypic deficiencies of PTEN knockout models may thus be partially attributed to loss of PTEN or one of its isoforms.

By use of specific disruption of PTENβ, our study reveals that PTENβ plays an essential role in rDNA transcription and regulation of cellular proliferation.

Several recent studies have described a diversity of nucleolar functions in addition to ribosome biogenesis, such as modification of small RNAs, regulation of the cell cycle and control of aging[47,48], and nucleolar perturbations have been observed in various diseases ranging from auto-immunity to cancer[49,50]. Our results show that in addition to nucleolin, PTENβ may physically interact with a number of other nucleolar proteins (Supplementary Fig. 4b), which raises the possibility that it participates in nucleolar functions apart from ribosome biogenesis.

In this study, we identify PTENβ as a new isoform of PTEN, which negatively regulates rDNA transcription and cellular proliferation. Based on previous studies that show imbalance of rDNA transcription closely correlates with tumour progression[51,52], identification of PTENβ may provide a new track for therapeutic drug design. These results together with our previous observations regarding PTENα illustrate the complexity of PTEN function and enrich our understanding of the PTEN family.

## Methods

**Cell lines**. All human cell lines used in this study (HepG2, H1299, HCT116, PC3, 786-O, Hela, U2OS, HEK 293T, SMMC-7721, DU145 and LoVo) were from the American Type Culture Collection. These cell lines were authenticated by STR locus analysis and were tested for mycoplasma contamination. PC3 and 786-O cells were maintained in RPMI 1640 (CORNING), and all other cells were maintained in DMEM (CORNING), supplemented with 10% FBS (Hyclone) in a 37 °C incubator with 5% (v/v) $CO_2$. The insect cell line Sf9 was obtained from Invitrogen and cultured in Grace's insect medium (Gibco).

**Mice**. Tissues of C57BL/6 mice[25] were lysed in lysis buffer (50 mM Tris-pH 7.5, 150 mM NaCl, 1% NP40) and subjected to immunoblotting or immunoprecipitation. All animals were maintained in a special pathogen-free facility, and the animal study protocols used were approved by the ethics committee of Peking University Health Science Center (approval number bjmu20110301).

**Antibodies and reagents**. Primary antibodies used for western blot, immunoprecipitation and immunofluorescence are listed in Supplementary Table 1. The anti-PTENα (ref. 25) and anti-nucleolin antibodies were raised in our laboratory (Supplementary Fig. 10). The reagent Acriflavin was from Sigma, A8126.

**Plasmids and cloning strategies**. The plasmids pCMV-tag-2b, pSA, pEGFP-N1, pDsRed2-N1 and pFastBac1 were purchased from Addgene. pX330 was purchased from BEIJING IDMO CO., LTD. pcDNA3.1-Cas9 was a gift from Dr Jianzhong Xi in the College of Engineering, Peking University. PTEN, PTENα, PTENβ, or nucleolin or truncations and mutations of these molecules were inserted into these plasmids. pHrD-IRES-Luc was a gift from Dr Xiaojuan Du in the Department of Cell Biology, Peking University Health Science Center[53].

**Figure 5 | PTENβ interacts with and dephosphorylates nucleolin. (a)** *In vivo* S-tag pull-down analysis. Whole cell extracts from HEK 293T cells transfected with S-tagged-PTEN, S-tagged-PTENα, S-tagged-PTENβ or mock constructs were immunoprecipitated with S-protein beads followed by mass spectrometric peptide sequencing. Nucleolin was found in the PTENβ pull-down list. **(b,c)** Reciprocal immunoprecipitation of PTENβ and nucleolin. **(b)** C-terminal GFP-tagged PTENβ or GFP-tagged mock was transfected, followed by immunoprecipitation with GFP antibody and immunoblotting with an anti-nucleolin antibody. **(c)** FLAG-tagged nucleolin or mock plasmid was co-transfected with GFP-tagged PTENβ, followed by immunoprecipitation with FLAG antibody and immunoblotting with GFP antibody. **(d)** Images of immunofluorescence staining for PTENβ and nucleolin. Hela PTEN null cells were transfected as in Fig. 4b, followed by staining with an anti-nucleolin antibody, and were imaged by confocal microscopy. The scale bars represent 5 μm. **(e)** PTENβ protein phosphatase activity is required for its interaction with nucleolin. Wild-type PTENβ or PTENβ mutants as indicated in Supplementary Fig. 6b was transfected separately, followed by immunoprecipitation with GFP antibody and immunoblotting with an anti-nucleolin antibody. **(f)** Schematic strategy of PTENβ CRISPR-Cas9 knockout. The initiation codons of PTEN isoforms were labelled. **(g)** Immunoblot confirming abolition of PTENα only (lane 2), or PTENα and PTENβ double knockout (lane 3) with a PTEN antibody. **(h)** Phospho-nucleolin level in PTENβ knockout cells. PTENβ knockout cells as indicated in **f,g** were lysed, followed by immunoblotting with p-nucleolin (Thr 84), nucleolin or β-actin antibody. Experiments were carried out in three biological replicates and quantification of indicated protein levels relative to β-actin is shown. Bars represent mean ± s.e.m. **(i)** C-terminal GFP-tagged wild-type PTENβ or PTENβ mutants as indicated in **e** were introduced into PTENα and PTENβ double knockout cells as indicated in **g**, followed by immunoblotting with p-nucleolin (Thr 84), nucleolin or GFP antibody. **(j)** Purified nucleolin was phosphorylated *in vitro* by Cdk1/CyclinB1 and subsequently used in a phosphatase assay with FLAG-PTENβ or FLAG peptide alone as a control, followed by immunoblotting with p-nucleolin (Thr 84).

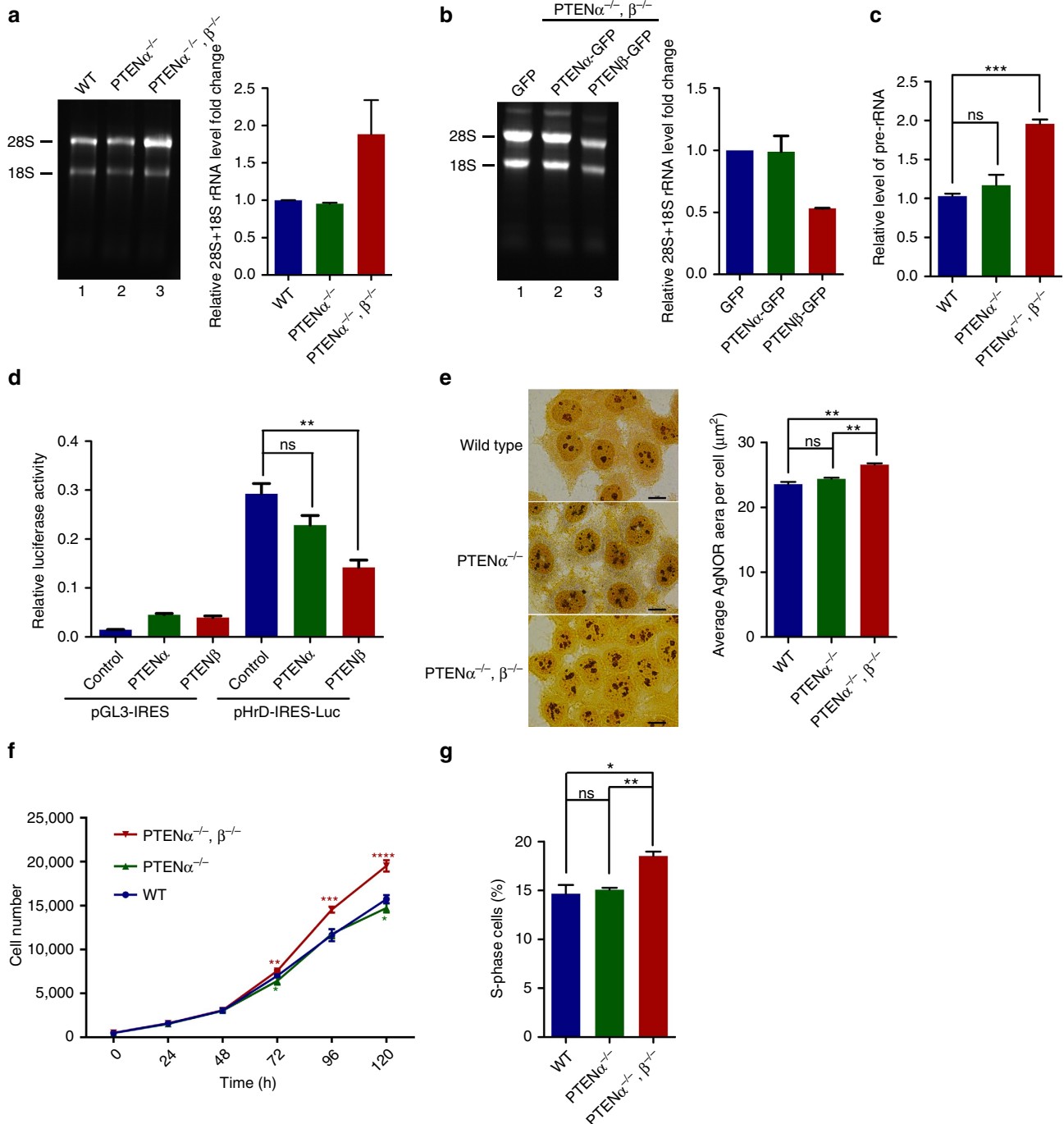

**Figure 6 | PTENβ negatively regulates rDNA transcription and cell proliferation.** (**a,b**) 28S and 18S rRNA levels in PTENβ knockout cells. (**a**) Total RNA was extracted from wild-type Hela cells and knockout cells as indicated in Fig. 5g, followed by resolution on a 1% formaldehyde-agarose gel for comparison. (**b**) Exogenous PTENα or PTENβ was transfected into PTENα and PTENβ double knockout cells as indicated in Fig. 5g, followed by resolution on a 1% formaldehyde-agarose gel for comparison. Grey scale quantification results of total 28S rRNA and 18S rRNA levels are shown. (**c**) 45S pre-rRNA level in PTENβ knockout cells as indicated in Fig. 5g. (**d**) rDNA promoter luciferase reporter activity in HeLa cells. PTENα and PTENβ double knockout cells as indicated in Fig. 5g were co-transfected with pHrD-IRES-Luc or pGL3-basic plasmid and exogenous PTENα or PTENβ. Luciferase activity was measured 24 h after transfection. (**e**) Average area of nucleolar organizing regions in cells as indicated in Fig. 5g. Left panel, images of silver staining. Right panel, quantitative analysis of AgNOR indices. AgNOR from 50 cells were measured in each group. The scale bars represent 10 μm. (**f**) Cell growth curves in cells as indicated in Fig. 5g. One thousand cells were plated per well in 96-well plates in triplicate. Quantification was performed every 24 h for 5 days following the instructions for the Cell Titer 96 AQueous One Solution Cell Proliferation Assay Kit. The significance of the difference in PTENα and PTENβ double knockout and wild-type Hela cells is shown with red asterisks, and the significance of the difference in PTENα only knockout and wild-type cells is shown with green asterisks. (**g**) Detection of S-phase cells using the APC BrdU Flow Kit. Wild-type, PTENα and PTENβ double knockout or PTENα knockout only cells were treated with 1 mM BrdU, followed by staining with fluorochrome-conjugated anti-Brdu antibody and evaluation with flow cytometry. Data are presented as mean ± s.e.m. of three independent experiments and were analysed with the paired *t*-test. *$P < 0.05$; **$P < 0.01$; ***$P < 0.001$; ****$P < 0.0001$.

**Mass spectrometry.** For in vitro PTENβ identification experiments, the PTENβ coding sequence (CUG$^{513}$-UGA$^{2240}$, PTENα initiation codon CUG$^{513}$ mutated to CUC and canonical PTEN initiation codon AUG$^{1032}$ mutated to AUA) was subcloned with a C-terminal His-tag into a pFastBac1 vector for expression in Sf9 insect cells. His affinity chromatography was used for purification of PTENβ-His before SDS–PAGE resolution of a 70-kDa band. Immunoprecipitated proteins were digested in-gel by endoproteinase Glu-C (Promega) following the manufacturer's instructions. Peptides were separated by online reversed-phase nanoscale capillary liquid chromatography (Easy-nLC 1,000, Thermo Scientific). The data dependent mass spectra were acquired with the LTQ-Orbitrap Elite mass spectrometer (Thermo Scientific) equipped with a nanoelectrospray ion source (Thermo Scientific). Raw files were searched by Proteome Discoverer (Version 1.4.1.14, Thermo Scientific) against the UniProt Human database supplemented with N-terminal extended PTEN sequence (Supplementary Fig. 11a). Search parameters were set as follows: enzyme with semi-Glu-C; up to two missed cleavages; carbamidomethyl cysteine as fixed modification; methionine oxidation and peptide N-terminal acetylation as variable modifications. The false discovery rates were set at 0.01 (ref. 25). For endogenous PTENβ identification experiments, a FLAG-purified 70 kDa band (molecular weight lower than PTENα) from Pten$^{FLAG}$ tissues was treated in the same way as described above. Raw files were searched with Proteome Discoverer (Version 1.4.1.14, Thermo Scientific) against the UniProt Mouse database supplemented with the N-terminal extended PTEN sequence (Supplementary Fig. 11b). For peptide analysis of pull down experiments, proteins were digested by endoproteinase Trypsin (Promega) in gel following the manufacturer's instructions. The following process was the same as described above. Raw files were searched by Proteome Discoverer (Version 1.4.1.14, Thermo Scientific) against the UniProt Human database. Search parameters were set as follows: enzyme with Trypsin; up to two missed cleavages; carbamidomethyl cysteine as fixed modification; methionine oxidation as variable modification. The false discovery rates were set at 0.01.

**CRISPR-CAS9-mediated somatic PTENβ knockout.** Sequences (ACCTCCCGCTCCTGGAGCGG or AGTCGCCTGTCACCATTTCC) of the human PTEN gene in exon1 were targeted separately. Oligos were purchased from Tsingke and ligated into the U6-sgRNA plasmid. Hela cells were seeded onto 24-well plates containing 500 μl DMEM medium at a density of 1,00,000 cells per ml of cell suspension. 200 ng of U6-sgRNA plasmid and 200 ng of Cas9 plasmid in total were transfected into cells using PEI according to the manufacturer's instructions. Medium was replaced at around 12 h post transfection. Cells were maintained for another 72 h to allow sufficient time for genomic engineering mediated by the CRISPR-Cas9 system. Transfected cells were then treated with 500 μg ml$^{-1}$ of G418 for 3 days. After G418 selection, cell clones were selected and amplified for mutation sequencing.

**Co-immunoprecipitation.** Immunoprecipitation was performed as previously described[12]. In brief, cells were extracted and lysed in lysis buffer (50 mM Tris-pH 7.5, 150 mM NaCl, 0.5% NP40) freshly supplemented with 1 mM PMSF and protease inhibitor. Cell lysate (500 μg) was incubated with antibody against GFP-tag or FLAG-tag for 8 h, followed by incubation with protein A/G agarose for 1 h. The protein-bead complex mixture was washed in washing buffer containing 0.1% NP40 and subjected to western blot to evaluate protein interaction.

**Immunofluorescence and confocal microscopy.** HeLa cells were seeded on cover glass in DMEM medium followed by transfection with C-terminal GFP-tagged PTEN, PTENα or PTENβ. After transfection for 24 h, cells were fixed with 4% PFA, permeabilized with 0.2% Triton X-100 and blocked for at least 30 min using 4% BSA dissolved in PBS + 0.1% Tween-20 (PBS-T). Primary antibody (diluted in blocking solution) was then applied for 1 h, followed by five washes in PBS-T, and fluorophore-conjugated secondary antibody was applied for 45 min. Coverslips were mounted and cells were evaluated with fluorescence microscopy. A Nikon A1 microscope was used for confocal microscopy.

**Cell fractionation.** Nucleoli were isolated in Hela cells as previously described with some modifications[54]. Isolated nucleoli were lysed with RIPA buffer. Equal amounts of whole cell lysate, cytosolic fraction, nuclear plasma fraction and nucleolar fraction were evaluated with western blot analysis. Briefly, Hela cells (one 15-cm dish) were scraped into ice-cold PBS and washed three times with ice-cold PBS. Cell pellets were then re-suspended in hypotonic buffer (10 mM Hepes, pH 7.9, 10 mM KCl, 1.5 mM MgCl$_2$, 0.5 mM DTT), allowed to swell and subjected to Dounce homogenization. The crude nuclear fraction was recovered by centrifugation at 218g for 5 min, and the cytoplasmic supernatant was retained. Nuclei were re-suspended in 0.35 M sucrose plus 0.5 mM magnesium acetate and then subjected to brief sonication to release the nucleoli. Pellets were purified by two cycles of centrifugation through 0.88 M sucrose, and the supernatant from the first sucrose gradient was recovered as a crude nucleoplasmic fraction. The pellets consisted of nucleoli.

**S-tag pull down assay.** HEK 293T cells were transfected with pSA, pSA-PTEN, pSA-PTENα, pSA-PTENβ or pSA-NCL plasmids and harvested 36 h after transfection. Cells were lysed with lysis buffer (50 mM Tris pH 7.5, 150 mM NaCl, 2 mM EDTA, 1% NP40 and 1 mM NaF). Equal amounts of protein were incubated with S protein agarose (Novagen) for 4 h. The protein-bead complexes were washed four times with washing buffer containing 0.1% NP40. After boiling at 100 °C in loading buffer, proteins were loaded onto NuPAGE 4–12% gels (Invitrogen) and visualized with silver staining (Pierce Silver Stain Kit) or were subjected to western blot. The potential interacting proteins in specific bands were evaluated with mass spectrum analysis.

**In vitro kinase and phosphatase assay.** His-nucleolin was expressed in Sf9 cells and purified using Ni-NTA agarose (Qiagen). Control FLAG peptides, FLAG-CDK1 and FLAG-PTENβ were purified with ANTI-FLAG M2 Affinity Gel (F2426) and 3 × FLAG Peptide (F4799) from HEK 293T cells after transfection with pCMV vector, pCMV-CDK1 or pCMV-PTENβ for 30 h. For kinase assay, purified His-nucleolin on Ni-NTA agarose was incubated with CDK1 in kinase buffer (50 mM Tris-HCl, 10 mM MgCl$_2$, 2 mM DTT, 1 mM EGTA and 0.2 mM ATP) for 30 min at 30 °C. For the following phosphatase assay, phosphorylated His-nucleolin was incubated with control FLAG peptide or purified FLAG-PTENβ in dephosphorylation buffer (20 mM HEPES, 1 mM MgCl$_2$, 1 mM EDTA, 1 mM DTT and 0.1 mg ml$^{-1}$ BSA) for 1 h at 37 °C. The phosphorylation levels of nucleolin sites with potential for alteration were evaluated with western blot.

**Functional pathway enrichment analysis and visualization.** Gene Ontology (GO) and KEGG enrichment analysis were performed using ClueGo (ref. 55), which is a Cytoscape plug-in. During analysis, over-represented terms in KEGG and the GO Biological Process and Cellular Component were enriched, and the GO level was set at three to five. The GO term fusion option was selected to reduce redundancy. In order to account for multiple testing, P values were corrected with Benjamini-Hochberg correction. Only terms of P < 0.05 were displayed. The selection criteria for the terms that have associated genes from the uploaded pull down list was set as a minimum of two genes per term, and a minimum of 4% of total genes associated with the term for the analysis of the PTENβ pull down network. The Kappa score was set at 0.4.

**Luciferase reporter assay and real-time PCR.** For luciferase reporter assays, after transfection of various combinations of control, reporter or protein expression vectors for 24 h, cells were lysed with a passive lysis buffer (Promega). Luciferase activity was measured with the Dual Luciferase Assay System (Promega) following the manufacturer's protocol. For quantification of 45S pre-rRNA, total RNA was extracted from cells followed by reverse transcription and evaluated with qPCR. The qPCR primers used are listed in Supplementary Table 4.

**Silver staining assay.** Silver staining assays were performed as previously described[56]. Briefly, cells were seeded onto glass coverslips overnight and were then fixed in 2% glutaraldehyde, followed by postfixation in a 3:1 ethanol–acetic acid solution. Cells were stained with a 0.33% formic acid–33.3% silver nitrate solution in 0.66% gelatin and mounted on slides with Vectashield (Vector Labs). Nucleolar area was measured in cells with cellSens software provided by Olympus.

**BrdU incorporation and cell cycle analysis.** For Brdu incorporation, cells were stained using the APC Brdu Flow Kit (BD Bioscience) according to the manufacturer's instructions and subjected to analysis using FACSVerse (BD Biosciences). Experiments were each performed three times, and at least 10$^4$ cells were analysed per sample.

**Statistical analysis.** Prism GraphPad software v5.0 was used for analysis. The statistical significance of differences between different groups was calculated with the two-tailed paired t-test. P values of 0.05 or less were considered significant.

**Data availability.** Nucleotide sequence data reported in this work are available in the Genebank databases under the accession numbers KX398936 (Homo sapiens) and KX421108 (Mus musculus). The mass spectrometry proteomics data have been deposited to the ProteomeXchange Consortium (http://proteomecentral.proteomexchange.org/cgi/GetDataset) via the PRIDE (ref. 57) partner repository with the data set identifier PXD003850. The original immunoblots, polyacrylamide gels and agarose gels are provided as Supplementary Figs 12–15. Primers used in this study are listed in Supplementary Tables 2–4. All other data supporting the findings of this study are available within the article and its Supplementary Information or from the corresponding author on reasonable request.

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

## Acknowledgements

We thank Dr Y. Jin for help with virus packaging, Dr L. Liang for help with Sf9 cell culture, Y. Liu for help with flow cytometry and Y. Li for help with mouse immunization.

This work was supported by grants to Y. Yin including the National Natural Science Foundation of China (Grants #81430056, #31420103905 and #81372491), the National Key Research and Development Program of China (Grant #2016YFA0500302), the Beijing Natural Science Foundation (Key Grant #7161007) and the Lam Chung Nin Foundation for Systems Biomedicine.

## Author contributions

Y.Y. and H.L. conceived the study. H.L. and Q.Y. wrote the article. H.L., X.C. and Q.Y. designed and performed the experiments and analysed the data. D.R. performed several experiments. X.Z. performed mass spectrometric analysis. C.Z. performed functional pathway enrichment analysis. Y.Y. and M.A.M. revised the manuscript.

## Additional information

**Competing financial interests:** The authors declare no competing financial interests.

