## [Peer Review File · Nature Communications]

Reviewers' comments:

Reviewer #1 (Remarks to the Author):

In this manuscript, Yuxin Yin and colleagues demonstrate the existence of yet another long form of PTEN in addition to the previously discovered PTEN_{long}/PTEN_α designated as PTEN_β. Given the existence of at least 2 PTEN_{long} isoforms, the authors appropriately suggest the use of sequential greek letters to designate these new isoforms. PTEN_β was first identified as a band migrating slightly faster than PTEN_α on western blots using different PTEN antibodies. Using various molecular and biochemical techniques the authors provide evidence not only for the existence of PTEN_β, but also for a distinct localization and function of this isoform compared to PTEN_α and canonical PTEN.

The work presented represents a novel discovery that may explain some of the pleiotropic cellular functions of the PTEN protein, in particular its previously reported role in ribosome function and translation. These findings will certainly generate interest in the fields of cancer biology, PTEN biology and translation/ribosome biology.

In general, the methodologies utilized in this study are appropriate and "state of the art" as expected from an established lab with renowned expertise in biochemical and molecular biology of PTEN. As such the authors have generated high quality data in most instances. Moreover the manuscript is very clearly written and polished (with only some minor typographical issues) and appropriately referenced.

Overall the authors provide a compelling argument for the existence of PTEN_β. However a few gaps in the data remain.

1. Notwithstanding the mass spec data performed on overexpressed PTEN_β to identify the initiation codon of PTEN_β is identified by MALDI-TOF MS, the manuscript lacks clear direct evidence to eliminate the possibility that the "alleged" PTEN_β may be some other protein species. The authors certainly have the expertise and tools to isolate endogenous PTEN_β (from cancer cell lines and or FLAG-Pten mouse cells) and confirm its identity by MS. In my opinion this readily obtainable data would be indisputable.

2. Regarding to the cellular localization of PTEN isoforms.

Although the data are convincing with regards to the respective localization of PTEN isoforms, it is not clear if the 5' extensions alone are driving the "completely different" PTEN localization among isoforms. Some possible experiments to address this may include...

- Are the 5'-extensions of alpha and beta solely responsible for the localization of PTEN isoforms. i.e if bound to a fluorescent protein, can the 5'-extensions of alpha and beta direct localization to the mitochondria and nucleolus, respectively.
- Are the localizations of PTEN, PTEN_α and PTEN_β mutually exclusive and thus mediating unique functions? By using different colored C-terminal fluorescent tags on the PTEN isoforms this could be effectively addressed in part.

3. Regarding the finding that "PTEN_α and PTEN_β double knockout led to elevation of nucleolin phosphorylation levels at Thr84, which is one of the XTPXKKXX motifs located in the nucleolin N-terminal region that is important for nucleolin function"

Despite the strong data supporting the interaction of PTEN_β with nucleolin, the conclusion that PTEN_β is a nucleolin protein phosphatase remains preliminary and thus still unfounded.

- The knockdown data (Fig 5g) should be complemented by "rescue" experiments
- overexpression of wild type PTEN_β and the specific phosphatase mutants presented in Fig 5d should further validate this finding

In my opinion, at least these experiments are required to make the conclusion about protein phosphatase activity of PTEN β on nucleolin.

Other Minor Comments:

- Line 132. Typo UUG622 ?

- Fig. 1b. Can you comment on the lack of PTEN α/β in H1299 cell line? It peculiarly stands out as the only line lacking PTEN α/β .

- Figure 2d. What is the 130kDa band? What are the bands migrating just above 100kDa...PTEN γ ? PTEN δ ?

- Line 227. There is no evidence that the "activity" of PTEN is essential for interaction with nucleolin. The only conclusion that can be made from you data is that the PTEN mutations alters the affinity for nucleolin binding.

Reviewer #2 (Remarks to the Author):

Liang_etal_review_082516

The authors have explored the biology of an alternate isoform of PTEN. Much of the evidence for the novel splice variant is supported by mass spectrometry data, which needs to be more fully explained. Furthermore, the raw data should be made available to the community in accordance with publication standards (for an example, see the guidelines from Molecular and Cellular Proteomics). The biological investigation is substantial, but some clarifications and possibly some additional data are needed to better support the protein sequence identification.

Line 50: de-repression seems awkward; why not just uncontrolled activation?

Line 92: acriflavin used twice in 3 words. Please revise.

Line 115 and Line 121 indicate different molecular weights for PTEN beta 70 and 120 kDa. Please revise for consistency.

Lines 156-160 Additional detail is required for the LC-MS/MS experiment. What type of proteolytic digestion is used? Trypsin digestion is typically used as the standard method; data from tryptic digests should be presented. See additional comments below under Figure 3.

Lines 227-230 Is the nucleolin phosphatase of PTEN beta more linked to its localization or is it a specific function of PTEN beta?

Line 233 and elsewhere in manuscript: Correct CRISPER to CRISPR.

Line 235-242 Is the nucleolin phosphatase activity of PTEN beta direct or indirect?

Line 249 Is there mass spectrometry data for dephosphorylation of Thr84 in nucleolin or immunoblotting? Please revise manuscript to include more detail on this point.

Line 365-366 PTEN beta is said to provide a new target for therapeutic drug design, which is hard to understand because the molecule is anti-tumor and anti-proliferation. Please provide additional clarification on this point.

Line 405 Additional brief detail is needed on the protein identification experiments, because neither this manuscript nor the reference has sufficient detail. Include method of proteolysis, MS instrument, protein database/version, database search engine, and criteria for positive peptide identification.

Line 451: Be more specific about mass spectrometry methods as in the previous query.

Lines 646-647 Similar to deposition of gene sequences, protein data should also be uploaded to a publically available database (e.g. PRIDE/ProteomeXchange).

Figure 3: Panel B has data that appear to be from a no enzyme database search; some Xcorr

values are too low to be acceptable. Figure 3C is too small to be legible; the manuscript reviewers and eventual journal readership are not able to evaluate this peptide identification.

Figure 4: Panel D is out of order in display; Panels B and C are difficult to see in the review copy.

Reviewer #3 (Remarks to the Author):

Liang, et al. report the identification of a novel, alternatively translated variant of PTEN, PTEN β . The authors show that PTEN β localizes to the nucleolus and plays a role in the regulation of ribosomal DNA transcription. Consistent with other findings that PTEN acts as a tumor suppressor, the authors show that disruption of PTEN β leads to increased ribosome biogenesis. However, in general the results as presented are sub standard and poorly controlled. In addition, the figures are not well put together, suffering from multiple examples of illegibility. I have enumerated multiple ways in which this manuscript could be improved.

Major Comments

1. Does disruption of the palindromic motif also lead to a reduction in PTEN expression?
2. Mass spectrometry analysis shows that PTEN β also interacts with POLR1C and POLR1D, components of the RNA polymerase I machinery that is responsible for ribosomal DNA transcription. While the effects of interaction with nucleolin are interesting, there should be analysis of the interaction with the transcription machinery itself, as this is likely a direct effect of PTEN β on ribosome biogenesis.
3. The discussion on PTEN's known role in ribosome biogenesis should be expanded. The only paper referenced is Podyspanina et al., but others have also investigated this. For example, Li, P., Wang, D., Li, H. et al. *Mol Biol Rep* (2014) 41: 6383. doi:10.1007/s11033-014-3518-6 and Cheng Zhang, Lucio Comai, and Deborah L. Johnson *Mol. Cell. Biol.* (2005) 25:16 6899-6911; doi:10.1128/MCB.25.16.6899-6911.2005.
4. For the luciferase assays, how many replicates were performed? Were the replicates biological or technical?
5. How many cells were counted for the silver staining assay?
6. In Figure 5d, which of the mutants are you suggesting abolishes phosphatase activity? Supplemental Figure 4b looks like C270S and G275E abolish activity while Y284L does not. In Figure 5d, though, it looks like G275E still interacts with nucleolin, while C270S and Y284L do not. Please include a detailed explanation of the interpretation of these results in the text.
7. Figure 5g needs replicates and quantification of phosphatase activity before any conclusions can be drawn.
8. Figure 6a and 6b need replicates and quantitation.
9. In Supplemental Figure 1, the PTEN antibodies used in b and c look different. Part b looks as though PTEN is not nucleolar while part c looks like PTEN is nucleolar.

Figure Issues

1. For Figure 1b, PTEN null cell lines should be listed in the text and should be indicated in the figure. For example, is H1299 PTEN null? Additionally, is there any explanation for cell-line specific expression differences of PTEN α and PTEN β ?
2. Figure 2c, the numbers in the mutations are unreadable.
3. Figure 2g is way too small and too blurry. No information can be obtained from the entire upper portion.
4. In Figure 3b, the bottom portion is unreadable.
5. Figure 3c cannot be interpreted at all.
6. Figure 4d is shown before 4c, which is confusing to the reader.
7. The western blot shown in Figure 5f could be clearer. It is difficult to interpret whether PTEN β is present or not given the low intensity of the bands.
8. Supplemental Figure 2b is unreadable.
9. In the figure legend for 2g, there is only part of the sentence going from line 686 to line 687.

Minor Comments

1. The in-text description of Figure 1a states that Figure 1a shows that PTEN β exhibits an

expression pattern identical to PTEN α . However, Figure 1a does not show any expression pattern. In fact, this is not shown until Figure 3d, and so there is no evidence to support this statement at that point in the article.

2. It should be CRISPR, not CRISPER. See line 233, 750, 751, 757, etc.
3. Insert the Hopkins et al. citation into line 300.
4. Line 373 there should be a space between "and" and "786-O".

Point-by-Point Response

A. Point-by-point responses to the comments by Reviewer #1

In this manuscript, Yuxin Yin and colleagues demonstrate the existence of yet another long form of PTEN in addition to the previously discovered PTEN-long/PTEN α designated as PTEN β . Given the existence of at least 2 PTENlong isoforms, the authors appropriately suggest the use of sequential greek letters to designate these new isoforms. PTEN β was first identified as a band migrating slightly faster than PTEN α on western blots using different PTEN antibodies. Using various molecular and biochemical techniques the authors provide evidence not only for the existence of PTEN β , but also for a distinct localization and function of this isoform compared to PTEN α and canonical PTEN.

The work presented represents a novel discovery that may explain some of the pleiotropic cellular functions of the PTEN protein, in particular its previously reported role in ribosome function and translation. These findings will certainly generate interest in the fields of cancer biology, PTEN biology and translation/ribosome biology.

In general, the methodologies utilized in this study are appropriate and “state of the art” as expected from an established lab with renowned expertise in biochemical and molecular biology of PTEN. As such the authors have generated high quality data in most instances. Moreover the manuscript is very clearly written and polished (with only some minor typographical issues) and appropriately referenced.

Overall the authors provide a compelling argument for the existence of PTEN β . However a few gaps in the data remain.

We would first like to express our sincere thanks to the reviewer for the positive comments and affirmation of our study. We found the reviewer's comments to be very helpful and of value for improving the quality of our data and manuscript. Accordingly, we have addressed each of the comments as follows.

Q1.1. *Notwithstanding the mass spectrometry data performed on overexpressed PTEN β to identify the initiation codon of PTEN β is identified by MALDI-TOF MS, the manuscript lacks clear direct evidence to eliminate the possibility that the “alleged” PTEN β may be some other protein species. The authors certainly have the expertise and tools to isolate endogenous PTEN β (from cancer cell lines and or FLAG-Pten mouse cells) and confirm its identity by MS. In my opinion this readily obtainable data would be indisputable.*

R1.1. We agree that confirmation of endogenous PTEN β by mass spectrometric analysis would be indisputable. As suggested, brain tissue samples were extracted from heterozygous *Pten*^{FLAG} mice and wild-type control mice for FLAG pull-down, followed by SDS-PAGE and silver staining for protein detection. A protein band with a molecular weight lower than PTEN α which was similar to the PTEN β band identified in cancer cell lines was detected in *Pten*^{FLAG} tissues, while no band was detected in wild-type control tissues. This PTEN β -like protein band was isolated and subjected to protein sequence identification with mass spectrometry. As shown in Fig. 3e, mascot reports revealed 4 peptide fragments that cover 68.5% of the N-terminal region of PTEN β . These results verify the existence of PTEN β *in vivo*.

Fig. 3e. Mass spectrometry analysis of endogenous PTEN β . 4 Peptide fragments that cover 68.5% of the N-terminal region of PTEN β are shown.

Q1.2. Regarding to the cellular localization of PTEN isoforms. Although the data are convincing with regards to the respective localization of PTEN isoforms, it is not clear if the 5' extensions alone are driving the "completely different" PTEN localization among isoforms. Some possible experiments to address this may include...

a) Are the 5'-extensions of alpha and beta solely responsible for the localization of PTEN isoforms. i.e if bound to a fluorescent protein, can the 5'-extensions of alpha and beta direct localization to the mitochondria and nucleolus, respectively.

b) Are the localizations of PTEN, PTEN α and PTEN β mutually exclusive and thus mediating unique functions? By using different colored C-terminal fluorescent tags on the PTEN isoforms this could be effectively addressed in part.

R1.2. According to these suggestions, the following experiments were carried out to address the role played by the N-terminal extended sequences in localization of PTEN isoforms.

- a) Plasmids expressing N-terminal extended sequences of PTEN α (AA 1-173) and PTEN β (AA1-146) were constructed with a C-terminal GFP tag. The indicated plasmids were introduced in HeLa PTEN null cells, followed by imaging with confocal microscopy. UBF was used as a nucleolar marker and MitoTracker was used to label mitochondria. Immunofluorescence revealed substantial co-localization of PTEN β (AA1-146) with UBF, while PTEN α (AA1-173) mainly localized on mitochondria (Supplementary Fig. 2). The N-terminal extended sequences are therefore solely responsible for the localization of PTEN isoforms.

Supplementary Fig. 2. The localization of N-terminal extended sequences of PTEN α (AA 1-173) and PTEN β (AA1-146). This shows that the N-terminal extended sequences are solely responsible for PTEN α localization on mitochondrion, and PTEN β accumulation in the nucleolus.

- b) Plasmids expressing C-terminal DsRed2 tagged PTEN α , C-terminal GFP tagged PTEN β and C-terminal FLAG tagged PTEN were constructed and co-transfected in HeLa PTEN null cells. A monoclonal FLAG antibody (Sigma-Aldrich, F3165) was used to label PTEN. Empty PEGFP-N1 and pDsRed2-N1 plasmids were transfected as controls. We found partial co-localization of PTEN and PTEN α in the cytoplasm which is consistent with the finding in our previous study showing a part of PTEN interacts and co-localizes with PTEN α on mitochondria¹. However, this exception aside, most of the PTEN, PTEN α and PTEN β showed mutually exclusive localization. PTEN was distributed ubiquitously in both the nucleus and cytoplasm, while PTEN α predominately showed cytoplasmic localization. In contrast, strong PTEN β fluorescence signals were distributed mainly in the relatively shallow DAPI staining areas of the nucleus, which have been suggested to be nucleolar regions (Supplementary Fig. 9). Thus, in agreement with the reviewer, we think these finding raises the possibility that PTEN isoforms may mediate unique functions based on distinct localization.

Supplementary Fig. 9. The majority PTEN isoforms show mutually exclusive localization.

Q1.3. *Regarding the finding that “PTEN α and PTEN β double knockout led to elevation of nucleolin phosphorylation levels at Thr84, which is one of the XTPXKKXX motifs located in the nucleolin. N-terminal region that is important for nucleolin function”. Despite the strong data supporting the interaction of PTEN β with nucleolin, the conclusion that PTEN β is a nucleolin protein phosphatase remains preliminary and thus still unfounded.*

The knockdown data (Fig 5g) should be complemented by “rescue” experiments. Overexpression of wild type PTEN β and the specific phosphatase mutants presented in Fig 5d should further validate this finding

In my opinion, at least these experiments are required to make the conclusion about protein phosphatase activity of PTEN β on nucleolin.

R.1.3. We agree that rescue experiments are required for drawing this conclusion and further validating PTEN β protein phosphatase activity on nucleolin. In addition, these comments inspired a more detailed mechanistic exploration to show nucleolin is a phosphatase substrate for PTEN β .

We thus introduced wild type PTEN β or phosphatase activity abolished PTEN β mutants (C270S, dual phosphatase activity abolished mutant, G275E, lipid phosphatase activity loss and Y284L, protein phosphatase activity loss) into PTEN α and PTEN β double knockout cells. We found that exogenous expression of wild type PTEN β , but not protein phosphatase activity abolished PTEN β (Y284L or C270S), effectively down-regulated nucleolin phosphorylation levels at Thr84 (Fig. 5i). These results help to confirm PTEN β acts as a nucleolin phosphatase.

Fig. 5i. The effect of PTEN β phosphatase activity abolished mutants on nucleolin phosphorylation levels at Thr84.

To further verify this point, *in vitro* dephosphorylation assays were carried out with purified PTEN β and nucleolin proteins. As shown in Fig. 5j, PTEN β efficiently down-regulated nucleolin phosphorylation levels at Thr84 *in vitro*. Taken together, these results argue that PTEN β acts as a nucleolin phosphatase.

Fig. 5j. *In vitro* dephosphorylation assays to evaluate the protein phosphatase activity of PTEN β on nucleolin.

Other Minor Comments

Q1.4. Line 132. Typo UUG⁶²² ?

R1.4. We are grateful to the reviewer for such careful review of our work, and we regret this labeling error. In the new revision, we have corrected this mistake "UUG⁶²²" to "UUG⁶²¹".

Q1.5. Figure 1b. Can you comment on the lack of PTEN α/β in H1299 cell line? It peculiarly stands out as the only line lacking PTEN α/β .

R1.5. The following efforts were made to address this concern. First, we attempted to determine whether lack of PTEN α/β signal in H1299 was due to absence of these proteins in this cell line, or whether detection of PTEN α/β failed simply due to lower protein expression in H1299, and insufficient use of protein lysate. To illustrate this point, we have performed the experiment by increasing the amount of total protein lysate, and the Hela cell line which has higher PTEN α/β levels was used as a positive control. It turned out that PTEN α/β was detectable in H1299 only when the total protein amount reached more than 100ug. The PTEN α/β level detected in 200ug H1299 protein lysate is equivalent to that in 50ug of Hela protein lysate (Fig. R1.1). We therefore conclude H1299 has a lower PTEN α/β level than other cell lines. This subsequently led to speculation that the H1299 cell line may harbor mutation(s) in the PTEN gene which affect expression or stability of PTEN α/β . To test this possibility, we extracted DNA from H1299 cells and sequenced the PTEN gene, but no PTEN mutation was identified. As shown in Fig. 3d, the expression level of PTEN α/β is variable in different tissues. It is noteworthy that the protein level of PTEN α/β is comparatively lower in lung than in most of the other tissues. In view of the fact that H1299 was originally derived from lung tissue, we speculate that the tissue-specific expression of PTEN α/β lead to lower levels of these proteins in the H1299 cell line.

Q1.6. Figure 2d. What is the 130kDa band? What are the bands migrating just above 100kDa...PTEN γ ? PTEN δ ?

R1.6. We thank the reviewer for these insightful comments.

a) We think the bands at about 130kDa molecular weight recognized by GFP antibody are

likely to be non-specific bands, because these bands can also be detected by GFP antibody in the control group which has been transfected with empty PEGFP-N plasmid only (Figure 2d, lane1).

- b) We agree with the reviewer that the bands migrating just above 100kDa might be other unidentified PTEN isoforms. These bands were also detected in Figure 2b. At the outset, we excluded the possibility these bands were non-specific bands detected by GFP antibody or PTEN modifications, as these specific bands cannot be detected in the group transfected with exogenous PTEN (Figure 2d, lane 2& Figure 2b, lane 3). As shown in Figure 2a, in addition to the initiation sites for PTEN α and PTEN β (CUG⁵¹³, AUU⁵⁹⁴), we found another two alternative non-AUG codons in favorable Kozak contexts which may be sites for initiation of other as yet unidentified isoforms. In particular, the predicted molecular weights of the proteins initiated from these two non-AUG codons with a GFP tag are about 100kDa, which is consistent with the bands detected in Figure 2d and Figure 2b. The bands migrating just above 100kDa in Figure 2d may therefore, be unidentified PTEN isoforms, which we intent to investigate in future studies.

Q1.7. Line 227. There is no evidence that the "activity" of PTEN is essential for interaction with nucleolin. The only conclusion that can be made from you data is that the PTEN mutations alters the affinity for nucleolin binding.

R1.7. We agree that the statement which is referred to is inaccurate. Accordingly, in the new revision the sentence "...we found that it is the protein phosphatase activity rather than the lipid phosphatase activity of PTEN β that is essential for interaction with nucleolin..." has been changed to "...we found that affinity for nucleolin binding was largely diminished in protein phosphatase activity deficient PTEN β mutants (Y284L and C270S)...".

B. Point-by-point responses to the comments by Reviewer #2

The authors have explored the biology of an alternate isoform of PTEN. Much of the evidence for the novel splice variant is supported by mass spectrometry data, which needs to be more fully explained. Furthermore, the raw data should be made available to the community in accordance with publication standards (for an example, see the guidelines from Molecular and Cellular Proteomics). The biological investigation is substantial, but some clarifications and possibly some additional data are needed to better support the protein sequence identification.

We really appreciate the reviewer's professional and constructive comments. In the new revision, our data has been reorganized in accordance with publication standards as suggested, and the mass spectrometry data has been more fully explained. Some new data has also been added to better support the protein sequence identification. We have addressed each comment, and details are listed as follows.

Q2.1. Line 50: de-repression seems awkward; why not just uncontrolled activation?

R2.1. We appreciate the reviewer's careful reading of our manuscript. We agree that "de-repression" is awkward, and this has now been changed to "uncontrolled activation" as suggested in the new revision.

Q2.2. Line 92: acriflavin used twice in 3 words. Please revise.

R2.2 As suggested, "...the CUG inhibitor acriflavin showed acriflavin reduces PTEN α expression..." has been changed to "The CUG inhibitor acriflavin reduces PTEN α expression in a dose-dependent manner without affecting canonical PTEN expression."

Q2.3. Line 115 and Line 121 indicate different molecular weights PTEN beta 70 and 120 kDa. Please revise for consistency.

R2.3. We regret any confusion caused by our inaccurate statement. The PTEN β described in line 121 (line 119 in the revised version) is an exogenously overexpressed PTEN β with a GFP tag. Thus, it has higher molecular weight than the endogenous PTEN β described in line 115 (line 113 in the revised version). Accordingly, in the new revision the description has been revised to avoid confusion.

Q2.4. Lines 156-160 Additional detail is required for the LC-MS/MS experiment. What type of proteolytic digestion is used? Trypsin digestion is typically used as the standard method; data from tryptic digests should be presented. See additional comments below under Figure 3.

R2.4. As suggested, additional brief details regarding these protein identification experiments have been added into the methods section in the revised version. Proteins were digested in-gel by endoproteinase Glu-C (Promega) following the manufacturer's instructions. Peptides were separated by online reversed-phase nanoscale capillary liquid chromatography (Easy-nLC 1000, Thermo Scientific). The data dependent mass spectra were acquired with the LTQ-Orbitrap Elite mass spectrometer (Thermo Scientific) equipped with a nanoelectrospray ion source (Thermo Scientific). Raw files were searched by Proteome Discoverer (Version

1.4.1.14, Thermo Scientific) against the UniProt Human database supplemented with N-terminal extended PTEN sequence (Supplementary Fig. 11a). Search parameters were set as follows: enzyme with semi-Glu-C; up to two missed cleavages; carbamidomethyl cysteine as fixed modification; methionine oxidation and peptide N-terminal acetylation as variable modifications. The false discovery rates (FDR) were set at 0.01.

Q2.5. Lines 227-230 Is the nucleolin phosphatase of PTEN β more linked to its localization or is it a specific function of PTEN β ?

R2.5. Thank you for this insightful comment. In the course of this revision, we demonstrated that a series of poly-arginine residues in the N-terminal extended domain of PTEN β function as a NoLS, and direct PTEN β 's localization in the nucleolus. The poly-arginine sequence deleted mutant completely eliminated PTEN β nucleolar accumulation (Fig. 4e). To address the reviewer's comment, wild type and poly-arginine sequence deleted PTEN β were introduced into PTEN α and PTEN β double knockout cells to test their ability to dephosphorylate nucleolin individually. We found that the poly-arginine sequence deleted PTEN β was unable to dephosphorylate nucleolin (Supplementary Fig. 7). This verifies that the nucleolin phosphatase activity of PTEN β is linked to its localization in the nucleolus.

Fig. 4e. The poly-arginine sequence deleted mutant completely eliminated PTEN β nucleolar accumulation.

Supplementary Fig. 7. The poly-arginine sequence deleted PTEN β was unable to dephosphorylate nucleolin at Thr84.

Q2.6. Line 233 and elsewhere in manuscript: Correct CRISPER to CRISPR.

R2.6. In this revised version, “CRISPER” has been corrected to “CRISPR”.

Q2.7. Line 235-242 Is the nucleolin phosphatase activity of PTEN beta direct or indirect?

R2.7. We thank the reviewer for this question. Accordingly, *in vitro* dephosphorylation assays were carried out to address it. His-nucleolin was expressed in Sf9 cells and purified using Ni-NTA agarose. Control FLAG peptides, FLAG-PTEN β and FLAG-CDK1 (a known nucleolin kinase) were purified with ANTI-FLAG® M2 Affinity Gel (F2426) and 3 \times FLAG Peptide (F4799) from HEK 293T cells after transfection with pCMV vector, pCMV-CDK1 or pCMV-PTEN β . His-nucleolin was phosphorylated by FLAG-CDK1, followed by incubation with control FLAG peptide or purified FLAG-PTEN β in dephosphorylation buffer. The nucleolin phosphorylation level at Thr84 was evaluated with western blot. We demonstrated that nucleolin can be dephosphorylated by PTEN β at Thr84 site *in vitro* (Fig. 5j lane 5 vs lane 4), which further verifies that PTEN β nucleolin phosphatase activity is direct.

Fig. 5j. Nucleolin can be directly dephosphorylated by PTENβ.

Q2.8. Line 249 Is there mass spectrometry data for dephosphorylation of Thr84 in nucleolin or immunoblotting? Please revise manuscript to include more detail on this point.

R2.8. As shown in R2.7., *in vitro* dephosphorylation assays were carried out according to this suggestion. Dephosphorylation of Thr84 in nucleolin by PTENβ was validated by subsequent immunoblotting with an anti-nucleolin-phosphorylated (Thr84) antibody (Fig. 5i).

Q2.9. Line 365-366 PTENβ is said to provide a new target for therapeutic drug design, which is hard to understand because the molecule is anti-tumor and anti-proliferation. Please provide additional clarification on this point.

R2.9. We regret this inaccurate statement. In view of the fact PTENβ is both anti-tumor and anti-proliferation, “identification of PTENβ may provide a new target for therapeutic drug design” has been changed to “identification of PTENβ may provide a new track for therapeutic drug design”.

Q2.10. Line 405 Additional brief detail is needed on the protein identification experiments, because neither this manuscript nor the reference has sufficient detail. Include method of proteolysis, MS instrument, protein database/version, database search engine, and criteria for positive peptide identification.

R2.10. As we noted in the response to Q2.4., details regarding LC-MS/MS experiments are described in the new revision.

Q2.11. Line 451: Be more specific about mass spectrometry methods as in the previous query.

R2.11. According to this suggestion, specific mass spectrometry methods have been added during revision. Proteins were digested by endoproteinase Trypsin (Promega) in gel following the manufacturer's instructions. Peptides were separated by online reversed-phase nanoscale capillary liquid chromatography (Easy-nLC 1000, Thermo Scientific). The data dependent mass spectra were acquired with the LTQ-Orbitrap Elite mass spectrometer (Thermo Scientific) equipped with a nanoelectrospray ion source (Thermo Scientific). Raw files were searched by Proteome Discoverer (Version 1.4.1.14, Thermo Scientific) against the UniProt Human database. Searching parameters were set as follows: enzyme with Trypsin; up to two missed cleavages; carbamidomethyl cysteine as fixed modification; methionine oxidation as variable modifications. The false discovery rates (FDR) were set at 0.01.

Q2.12. Lines 646-647 Similar to deposition of gene sequences, protein data should also be uploaded to a publically available database (e.g. PRIDE/ProteomeXchange).

R2.12. As suggested, The mass spectrometry proteomics data have been deposited to the ProteomeXchange Consortium via the PRIDE² partner repository with the dataset identifier PXD005348. The reviewer account details for ProteomeXchange Consortium are as follows. Username: reviewer97789@ebi.ac.uk, password: oTjwBSlq.

Q2.13. Figure 3: Panel B has data that appear to be from a no enzyme database search; some Xcorr values are too low to be acceptable. Figure 3C is too small to be legible; the manuscript reviewers and eventual journal readership are not able to evaluate this peptide identification.

R2.13. We are grateful for these comments. Search parameters were set as follows: enzyme with semi-Glu-C; up to two missed cleavages; carbamidomethyl cysteine as fixed modification; methionine oxidation and peptide N-terminal acetylation as variable modifications. According to the reviewer's comment, the peptides with comparatively lower Xcorr values and the peptides which do not conform to search parameters were excluded. The MS/MS spectrum data have been enlarged to be more readable in the revised version.

Revised Fig. 3b. Mass spectrometry analysis of purified PTEN β protein.

Q2.14. Figure 4: Panel D is out of order in display; Panels B and C are difficult to see in the review copy.

R2.14. We regret the irregularity of this display noted by the reviewer, and this has been corrected in the revised version. As the reviewer emphasized that panel B and Panel C were difficult to see in the review copy, these data have been enlarged to be more readable as shown here.

Revised Fig. 4b. The localization of PTEN isoforms.

Revised Fig. 4c. The colocalization of PTEN isoforms with the nucleolar marker UBF.

C. Point-by-point responses to the comments by Reviewer #3

Liang, et al. report the identification of a novel, alternatively translated variant of PTEN, PTEN β . The authors show that PTEN β localizes to the nucleolus and plays a role in the regulation of ribosomal DNA transcription. Consistent with other findings that PTEN acts as a tumor suppressor, the authors show that disruption of PTEN β leads to increased ribosome biogenesis. However, in general the results as presented are sub standard and poorly controlled. In addition, the figures are not well put together, suffering from multiple examples of illegibility. I have enumerated multiple ways in which this manuscript could be improved.

We appreciate the reviewer's comments and instructions for improving our data and manuscript. In the revised version, results have been better presented to be in accordance with publication standards, and figures have been rearranged to repair illegibility. We have done an additional series of experiments and addressed all the comments as below:

Q3.1. *Does disruption of the palindromic motif also lead to a reduction in PTEN expression?*

R3.1. The initiation codon of canonical PTEN was mutated in the plasmids used in the original Fig. 2h lane3 and lane4, which was not clearly described. We regret any confusion caused by our inadequate statement. To clarify whether the disruption of the palindromic motif, which is important for PTEN β initiation affects PTEN expression, we have redone the

experiments in the original Fig. 2h using plasmids which do not carry any mutation of the initiation codon of PTEN. We found that disruption of the palindromic motif led to a marked reduction of PTEN β , while PTEN expression was not affected. To avoid confusion, the original Fig. 2h has been replaced in the revised version.

Revised Fig. 2h. Disruption of the AUU⁵⁹⁴ downstream palindromic motif led to a marked reduction of PTEN β without affecting canonical PTEN expression.

Q3.2. Mass spectrometry analysis shows that PTEN β also interacts with POLR1C and POLR1D, components of the RNA polymerase I machinery that is responsible for ribosomal DNA transcription. While the effects of interaction with nucleolin are interesting, there should be analysis of the interaction with the transcription machinery itself, as this is likely a direct effect of PTEN β on ribosome biogenesis.

R3.2. To address this, immunoprecipitation experiments were carried out and nucleolin that had been demonstrated to interact with PTEN β (Fig. 5b) was used as a positive control. The interaction between PTEN β and nucleolin was again verified, while POLR1C and POLR1D were not detected as PTEN β interacting proteins (Fig. R3.1). Therefore it is unlikely that PTEN β exerts its function through direct interaction with POLR1C or POLR1D. Except for nucleolin, POLR1C and POLR1D, mass spectrometry analysis revealed that there are other potential PTEN β interacting nucleolar proteins involved in various nucleolar functions apart from rDNA transcription (Supplementary Fig. 4b). The interaction of these proteins with PTEN β needs further confirmation by immunoprecipitation, and we will undertake the aspect of the investigation in the future.

Fig. R3.1. Immunoprecipitation to detect the interaction between nucleolin, POLR1C or POLR1D and PTEN β . HEK 293T cells were transfected with C-terminal GFP tagged PTEN β or GFP tagged mock,

followed by immunoprecipitation with GFP antibody and western blotting with a nucleolin, POLR1C, or POLR1D antibody. Nucleolin was detected as a positive control. POLR1C and POLR1D were not detected as PTEN β interacting proteins.

Q3.3. *The discussion on PTEN's known role in ribosome biogenesis should be expanded. The only paper referenced is Podyspanina et al., but others have also investigated this. For example, Li, P., Wang, D., Li, H. et al. Mol Biol Rep (2014) 41: 6383. doi:10.1007/s11033-014-3518-6 and Cheng Zhang, Lucio Comai, and Deborah L. Johnson Mol. Cell. Biol. (2005) 25:16 6899-6911; doi:10.1128/MCB.25.16.6899-6911.2005.*

R3.3. According to the reviewer's comments, the discussion of PTEN's known role in ribosome biogenesis has been expanded in the revised version.

Q3.4. *For the luciferase assays, how many replicates were performed? Were the replicates biological or technical?*

R3.4. These luciferase assays were carried out in three biological replicates. For each biological replicate, four technical replicates were measured.

Q3.5. *How many cells were counted for the silver staining assay?*

R3.5. 50 cells were counted in each group in the silver staining assay. This information has been added in the revised version.

Q3.6. *In Figure 5d, which of the mutants are you suggesting abolishes phosphatase activity? Supplemental Figure 4b looks like C270S and G275E abolish activity while Y284L does not. In Figure 5d, though, it looks like G275E still interacts with nucleolin, while C270S and Y284L do not. Please include a detailed explanation of the interpretation of these results in the text.*

R3.6. In the PI3K/AKT pathway, PTEN converts PIP3 into PIP2 by its lipid phosphatase activity, which directly reverses the effect of PI3K and catalyzes its reaction in the opposite direction (PIP2 to PIP3). In turn, PIP3 is necessary for the phosphorylation of AKT to pAKT. Thus PTEN indirectly downregulates the pAKT level through its lipid phosphatase activity³.

PTEN β (G275E, analogous to PTEN (G129E)) is a lipid phosphatase activity abolished mutant; PTEN β (Y284L, analogous to PTEN (Y138L)) is a protein phosphatase activity abolished mutant; PTEN β (C270S, analogous to PTEN (C124S)) is a both lipid and protein phosphatase activity abolished mutant. Thus, in the original Supplementary Figure 4b (revised Supplementary Figure 6b), the pAKT level can be efficiently downregulated by wild type PTEN β and the PTEN β mutant (Y284L), and cannot be downregulated by the lipid phosphatase activity abolished PTEN β (G275E) and (C270S) mutants.

As to the original Fig. 5d (revised Fig. 5e), we found that the affinity for nucleolin binding of protein phosphatase activity abolished PTEN β (Y284L) or (C270S) was much weaker than wild type PTEN β . We thus speculated nucleolin may be a protein phosphatase substrate for PTEN β .

We regret any confusion caused by our inadequate statement. Accordingly, the PTEN β mutants have been fully described in the text and in the corresponding figure legends.

Q3.7. *Figure 5g needs replicates and quantification of phosphatase activity before any conclusions can be drawn.*

R3.7. In light of the reviewer's suggestion, we performed this experiment with three replicates, and quantitative analysis was carried out. As shown in the revised Fig. 5h, the results support our statement that double knockout of PTEN α and PTEN β should result in elevated phosphorylation of nucleolin at the Thr84 site, while knockout of PTEN α alone did not have this effect. This supports the view that PTEN β acts as a protein phosphatase for nucleolin.

Revised Fig. 5h. The effect of PTEN β knockout on phosphorylation level of nucleolin (Thr84).

Q3.8. *Figure 6a and 6b need replicates and quantitation.*

R3.8. As suggested, three replicates were performed and the results were quantitated. Gray scale quantification results of total 28S and 18S rRNA level are shown in the revised version to illustrate our statement.

Revised Fig. 6a (left panel) & 6b (right panel). Negative correlation of PTEN β and the total amount of 28S and 18S rRNA.

Q3.9. In Supplemental Figure 1, the PTEN antibodies used in b and c look different. Part b looks as PTEN is not nucleolar while part c looks like PTEN is nucleolar.

R3.9. We agree that in original Supplemental Figure 1, the nucleolar localization of PTEN looks different in b and c. To address this comment, we have redone this experiment. More than 50 cells with fluorescence signals were checked in each group. We found that there was no detection of PTEN fluorescence signal in the nucleolus in most cells, using the primary antibody PTEN (sc-7974) or PTEN (ABM-2502). In addition, we noticed that the nuclear morphology of the cell chosen to show PTEN localization in the original Supplementary Figure 1c is irregular compared with other cells, indicating that it is not representative. The image showing PTEN localization in original Supplementary Fig. 1c has therefore been replaced with one showing a typical cell.

Revised Supplementary Fig. 1c Localization of PTEN, PTEN α and PTEN β detected by immunofluorescence with PTEN (sc-7974) antibody.

Figure issues

Q3.10. For Figure 1b, PTEN null cell lines should be listed in the text and should be indicated in the figure. For example, is H1299 PTEN null? Additionally, is there any explanation for cell-line specific expression differences of PTEN α and PTEN β ?

R3.10. We thank the reviewer for the careful review of our data.

- a) The prostate cancer cell line PC3 is PTEN null and this has now been noted in the text in the revised version.
- b) As we mentioned in the response to Reviewer #1 Q1.5., the following efforts were made to address the possibility that H1299 is PTEN null. We first tried to determine whether H1299 lacks PTEN α/β , or whether there was a failure of PTEN α/β detection due to lower protein expression in H1299 and insufficient protein lysate (50ug) in the original

experiments. To evaluate this point, we have performed the experiment by increasing the amount of total protein lysate, and HeLa which has higher PTEN α/β level was used as a control. It turned out that it was only when the total protein amount reached more than 100ug that PTEN α/β was detectable in H1299 cells. The PTEN α/β level detected in 200ug of H1299 protein lysate is equivalent to that in 50ug of HeLa protein lysate (Fig. R1.1). We therefore conclude that H1299 has lower PTEN α/β levels than other cell lines.

- c) As to the cell-line specific expression differences of PTEN α and PTEN β , we speculated that the cell lines with lower PTEN α/β levels as shown in Fig. 1b may harbor mutation(s) in PTEN gene which affect expression or stability of PTEN α/β . To test this possibility, we extracted DNA from H1299, HEK 293T, DU145 and LoVo cell lines which have comparatively lower PTEN α/β levels, and sequenced the PTEN gene, but no PTEN mutation was identified. As shown in Fig. 3d., the expression level of PTEN α/β is variable in different tissues. It is noteworthy that the protein level of PTEN α/β is comparatively lower in lung and kidney than in most of the other tissues. In view of the fact H1299 and HEK 293T cell lines were derived from lung cancer tissue or embryonic kidney cells respectively, we speculate that tissue-specific expression of PTEN α/β leads to cell-line specific expression differences of PTEN α and PTEN β .

Q3.11. Figure 2c, the numbers in the mutations are unreadable.

R3.11. According to this comment, the numbers are now shown in larger fonts in the revised version.

Revised Fig. 2c. A set of constructs of PTEN and PTEN α with a C-terminal GFP tag, in which one of two possible sites (AUU⁵⁹⁴ or UUG⁶²¹) was mutated to CUC combined with mutation of CUG⁵¹³.

Q3.12. Figure 2g is way too small and too blurry. No information can be obtained from the entire upper portion.

R3.12. We apologize for this poor presentation of data. The figure noted has now been enlarged.

Revised Fig. 2g. The 5' UTR of PTEN containing a 12 bp AUU⁵⁹⁴ downstream palindromic motif which is evolutionarily conserved.

Q3.13. In Figure 3b, the bottom portion is unreadable.

R3.13. The data as noted has been enlarged and reorganized to be more readable in the revised version.

Revised Fig. 3b. Mass spectrometry analysis of purified exogenous PTEN β .

Q3.14. Figure 3c cannot be interpreted at all.

R3.14. As we noted in the response to Reviewer #2 Q2.13., the MS/MS spectrum data have been enlarged to be more readable in the revised version.

Revised Fig. 3c. The MS/MS spectrum of the peptide (MSRAGNAGE) that matches the N-terminal sequence of PTEN β .

Q3.15. Figure 4d is shown before 4c, which is confusing to the reader.

R3.15. We regret the irregularity of this display of data display as noted by the reviewer, and this has been corrected in the revised version.

Q3.16. The western blot shown in Figure 5f could be clearer. It is difficult to interpret whether PTEN β is present or not given the low intensity of the bands.

R3.16. We improved the immunoblotting by increasing the amount of protein lysate and consequently as shown in our revised Fig. 5g, we obtained clear bands for PTEN, PTEN α and PTEN β . As shown, PTEN α cannot be detected in PTEN α knockout cells, whereas both PTEN α and PTEN β were absent in double knockout cells (revised Fig. 5g, lane 2 vs lane 1; and lane 3 vs lane 1).

Revised Fig. 5g. Western blot confirming elimination of PTEN α only or PTEN α and PTEN β double knockout of somatic knockout cells induced with CRISPR-Cas9.

Q3.17. Supplemental Figure 2b is unreadable.

R3.17. According to this comment, the data noted (Supplementary Fig. 4b in the revised version) has been enlarged and reorganized to be more readable in the revised version.

Revised Supplementary Fig. 4b. Functionally grouped KEGG and GO Biological Process and Cellular Component term annotation network of PTEN β pull-down genes.

Q3.18. In the figure legend for 2g, there is only part of the sentence going from line 686 to line 687.

R3.18. We regret this error, and this has now been corrected in the revised version.

Minor comments

Q3.19. The in-text description of Figure 1a states that Figure 1a shows that PTEN β exhibits an expression pattern identical to PTEN α . However, Figure 1a does not show any expression pattern. In fact, this is not shown until Figure 3d, and so there is no evidence to support this statement at that point in the article.

R3.19. We thank the reviewer for this comment. We have quantified the ratio of PTEN α or PTEN β and GAPDH, and consequently confirmed that under acriflavin treatment the level of PTEN α and PTEN β declined in a parallel manner. To avoid confusion, the quantification result was added into the revised version, and the in-text description has been changed to “expression of this protein showed dose dependent variation under acriflavin treatment parallel to that of PTEN α ”.

Revised Fig. 1a. An unidentified protein recognized by PTEN antibody, the expression of which showed dose dependent variation under acriflavin treatment parallel to that of PTEN α . Quantification of indicated protein levels relative to GAPDH is shown (right panel).

Q3.20. It should be CRISPR, not CRISPER. See line 233, 750, 751, 757, etc.

R3.20. In this revised version, the misspelling “CRISPER” has been corrected to “CRISPR”.

Q3.21. Insert the Hopkins *et al.* citation into line 300.

R3.21. This citation has been inserted in the revised version.

Q3.22. Line 373 there should be a space between “and” and “786-O”.

R3.22. We are grateful for the reviewer’s careful reading of our manuscript. The space has been added accordingly.

References

1. Liang, H. *et al.* PTEN α , a PTEN isoform translated through alternative initiation, regulates mitochondrial function and energy metabolism. *Cell metabolism* **19**, 836-848 (2014).
2. Vizcaino, J.A. *et al.* 2016 update of the PRIDE database and its related tools. *Nucleic acids research* **44**, D447-456 (2016).
3. Yin, Y. & Shen, W.H. PTEN: a new guardian of the genome. *Oncogene* **27**, 5443-5453 (2008).

REVIEWERS' COMMENTS:

Reviewer #1 (Remarks to the Author):

No further comments. The authors have addressed each of the concerns I have raised.

Reviewer #2 (Remarks to the Author):

Thank for the responses. The changes have significantly improved the manuscript. No further revisions are requested.

Reviewer #3 (Remarks to the Author):

No further comments.